**Assesing the Value of High-Resolution Data and Parameters Transferability**
**Across Temporal Scales in Hydrological Modeling: A Case Study in Northern**
**China**
Mahmut Tudaji, Yi Nan*, Fuqiang Tian*
Department of Hydraulic Engineering & State Key Laboratory of Hydroscience and Engineering,
Tsinghua University, Beijing 100084, China
*Correspondence to*: Yi Nan (ny1209@qq.com), Fuqiang Tian (tianfq@tsinghua.edu.cn)
Abstract: The temporal resolution of input data and the computational time step are crucial factors
affecting the accuracy of hydrological model forecasts. This study presents a four-source hydrological
model tailored to the runoff characteristics of the mountainous areas in Northern China. Using this model,
along with meteorological and hydrological data from seven catchments of varying sizes in Northern
China, we investigated the impact of different input data resolutions and computational time steps on
simulation accuracy, as well as the transferability of parameters across different time scales. The results
show that: (1) The proposed model performs well across different spatial and temporal scales, with
average NSE for daily and hourly flow forecasts of 0.93 and 0.85, respectively. (2) For daily streamflow
simulations, there was a significant improvement in model performance when the data resolution was
increased from 24 hours to 12 hours; however, beyond the 12-hour resolution, the improvement became
negligible. For hourly streamflow simulations, the enhancement in overall flood process accuracy
becomes insignificant when the resolution exceeds 6 hours, although higher resolutions continue to
improve the precision of peak flow simulations. (3) When the computational time step is fixed (e.g., 1
hour), model parameters are transferable across different data resolutions; parameters calibrated with
daily data can be used in models driven by sub-daily data. However, parameters are not transferable when
the computational time step varies. Therefore, it is recommended to utilize smaller computational time
step when constructing hydrological models even in the absence of high-resolution input data. This
strategy ensures that the same simulation accuracy can be achieved while preserving the transferability
of model parameters, thus enhancing the robustness of the model.
**1 Introduction**
Hydrological modeling plays a critical role in water resources management, flood forecasting, and
climate impact assessments. Accurate simulation of runoff processes is essential for understanding water
balance and predicting hydrological extremes. The effectiveness of a hydrological model is influenced
by the scale of input data (resolution), the scale of the model's computation, and the scale of the
hydrological processes being modelled (López-Moreno et al., 2013; Merheb et al., 2016).
In the past, hydrological modeling has typically relied on daily or coarser resolution data, limiting its
applicability for shorter time steps required in scenarios like flash flood forecasting. Models that utilize
coarse or artificially enhanced data may introduce biases when applied to finer temporal scales, as they
may fail to accurately represent the variability and magnitude of key hydrological variables. However,
advancements in measurement technologies, including high-frequency automated rain/streamflow
gauges and phased array rain-radars, have enabled access to high-resolution rainfall and runoff datasets.
Despite these technological advances, the quantitative benefits of high-resolution data in enhancing
hydrological model performance remain unclear. For instance, studies on the impact of rainfall data
resolution on hydrological models have produced inconsistent results. Kobold and Brilly (2006) found
that calibrating hydrological models with sub-daily data and time steps can significantly improve the
accuracy of flood forecasting, comparing with using daily data and time steps. Jeong et al. (2011)
observed similar improvements. Huang et al. (2019) found that increasing spatial resolution has only a
marginal or minimal effect on model performance, while high temporal resolution data leads to a
significant improvement in model performance. However, other studies (Kannan et al., 2007; Ficchì et
al., 2016) have found that higher data resolution does not always lead to better model performance.
Ficchì et al. (2016) reported that as the time scale is reduced, the improvement in model performance
becomes limited, and performance may even degrade. Our previous research (Tudaji et al., 2025) in
southern China showed that high temporal resolution data does not always have positive impact on model
performance. Nevertheless, we and other related studies acknowledge that further studies across different
climate zones and models are necessary to validate and extend the generality of these findings.
Moreover, there remain other unresolved issues regarding data resolution that warrant further
investigation. When a certain resolution is selected for a watershed model based on current data
availability (or a specific standard) and the model's parameters are calibrated accordingly, the model is
essentially considered constructed. Nevertheless, should the resolution of subsequent input data deviate
from the one employed in the model's creation, the continued reliability of the model's predictive
outcomes becomes questionable. There is a need to explore whether the model's parameters were
optimized solely to maximize simulation metrics for that particular resolution, and whether these
parameters can be transferred effectively across different data resolutions. Furthermore, an in-depth
investigation into how model parameters adapt to input data of varying resolutions is instrumental in
uncovering the specific impacts of these data on hydrological simulation outcomes, as well as elucidating
the mechanistic aspects of how changes in resolution influence the hydrological simulation process.
Reynolds et al. (2017) found that the model calibrated by the daily data performance almost as good as
the model calibrated by data at sub-daily resolutions. However, this conclusion was reached under a fixed
computational time step, and the study (including the aforementioned studies on input data resolution)
also acknowledges that the generality of their conclusions to other regions and models warrants further
investigation.
Similarly, another issue that arises when constructing hydrological models is the choice of the model's
computational time step. The time dependence and transferability of parameters has been widely studied.
(Krajewski et al. 1991; Finnerty et al. 1997; Littlewood and Croke 2008; Reynolds et al. 2017). Recent
studies have provided quantitative insights into relationship of parameters at different computation time
steps. Wang et al. (2009) established the relationship between the parameters and the square root of the
time step; Jie et al. (2018) established transformation function between parameter values at different time
steps. However, it remains uncertain whether a finer computational time step consistently leads to
improved simulation accuracy when the resolutions of input and output are fixed. Moreover, the extent
to which parameters can be transferred across different computational time steps without transformation
and the existence of an optimal computational time step that maximizes both parameter transferability
and model performance are still questions that warrant further investigation.
In light of these background, this study seeks to enhance our understanding of the value of high-resolution
data and transferability of parameters across temporal scales in hydrological modeling in a new climatic
region using a new model. This study aims to complement studies on the effects of time scales in different

climate regions, and also provide explanations of the time-scale effect of models from a simulation process perspective, based on parameter variations. Seven small-to-medium catchments in northern China (a semi-humid and semi-arid region) were selected as the study area, with the aim of leveraging its unique climate and runoff characteristics to provide new insights and data analysis for hydrological modeling research and practice. We designed two experiments focusing on the most common hydrological forecasting timescales—daily and hourly, to investigate the value of the high-resolution data on hydrological modelling, using data resolutions ranging from 1 to 24 hours. Besides, two further experiments, one with various data resolutions and another with various computation time steps, were conducted to assess the transferability of parameters under different conditions. Specifically, this study seeks to address three key questions:

(1) What is the necessary resolution of rainfall and streamflow data to provide reliable hourly and daily streamflow simulations?

(2) When the computation time step is fixed as hourly, can parameters be transferred when adopting different temporal resolutions of input data?

(3) When the temporal resolution of input data is fixed as daily, can parameters be transferred when adopting different computation time steps?

The rest of this paper is structured as follows: Section 2 outlines the materials and methodology, including the introduction of study catchments, the hydrological model used, and the experimental designs. Section 3 presents the results of the experiments. Section 4 explains the role of high-resolution data, discusses the transferability of parameters under different conditions, and provides insights into selecting data resolution and computation time step during the modelling. Finally, Section 5 offers concluding remarks and limitations in this study.

## 2 Materials and methodology

### 2.1 Study area and data

The Chaobai River, located in northern China and flowing through Beijing, is one of the five major rivers in the Haihe River system of China. In this study, we utilized a set of 7 various size of catchments in the upper reaches of the Chaobai River as the study area (Figure1, Table1), where data quality is relatively high and human activities (such as reservoirs or dams) have minimal impact. Among them, the Xitaizi Basin, the smallest one, is a hydrological experimental catchment. The other six study catchments are the control regions of important hydrological stations located upstream of reservoirs or lakes on the major tributaries in the upper reaches of the Chaobai River Basin. Considering the existence of the Baihepu (BHP) Reservoir in the Baihe watershed, and to exclude human interference and the accumulation of simulation errors, the catchment area between the Baihepu Reservoir, THK, and ZJF was treated as an independent catchment. The measured outflow from the Baihepu Reservoir and the measured flow at THK are used as known boundary conditions in the hydrological model to simulate the flow at ZJF.

The study area is located in a semi-humid, semi-arid region, characterized by a temperate monsoon climate, with precipitation highly seasonal and primarily concentrated in July and August, resulting in significant seasonal and interannual variations in river flow. During periods outside the rainy season, the flow is minimal, and in some cases, the river may even run dry. Therefore, we chose the 2021 flood season, which saw significant flood events and has relatively complete data, as the study period for this study.

The streamflow and rainfall data were obtained from the Rain and Hydrological Database of Beijing, curated by the Beijing Hydrological Station. When selecting the above-mentioned hydrological stations as the outlets for the study basins, the following principles were followed: (1) The station must have discharge data with a resolution finer than hourly during flood events; (2) The upstream control area of the station should be free of water control structures such as reservoirs, dams, or lakes that could significantly affect the natural progression of floods; and (3) The study catchments should cover a range of different scales, from a few square kilometers to several thousand square kilometers. The selection of rainfall data followed similar principles, ensuring that each rain gauge station provided complete rainfall data with a resolution finer than hourly throughout the entire storm runoff process. We identified 56 high-quality stations situated within the study catchments from the database. The number of rainfall gauges per catchment varied from 1 to 14, averaging 8 stations. Additionally, the rainfall gauging area—calculated as the catchment area divided by the number of stations—ranged from 3 km² to 373 km², with an average of 157 km². The Thiessen Polygon method (Han and Bray, 2006) was employed to generate the areal rainfall data for each sub-basin in each catchment.

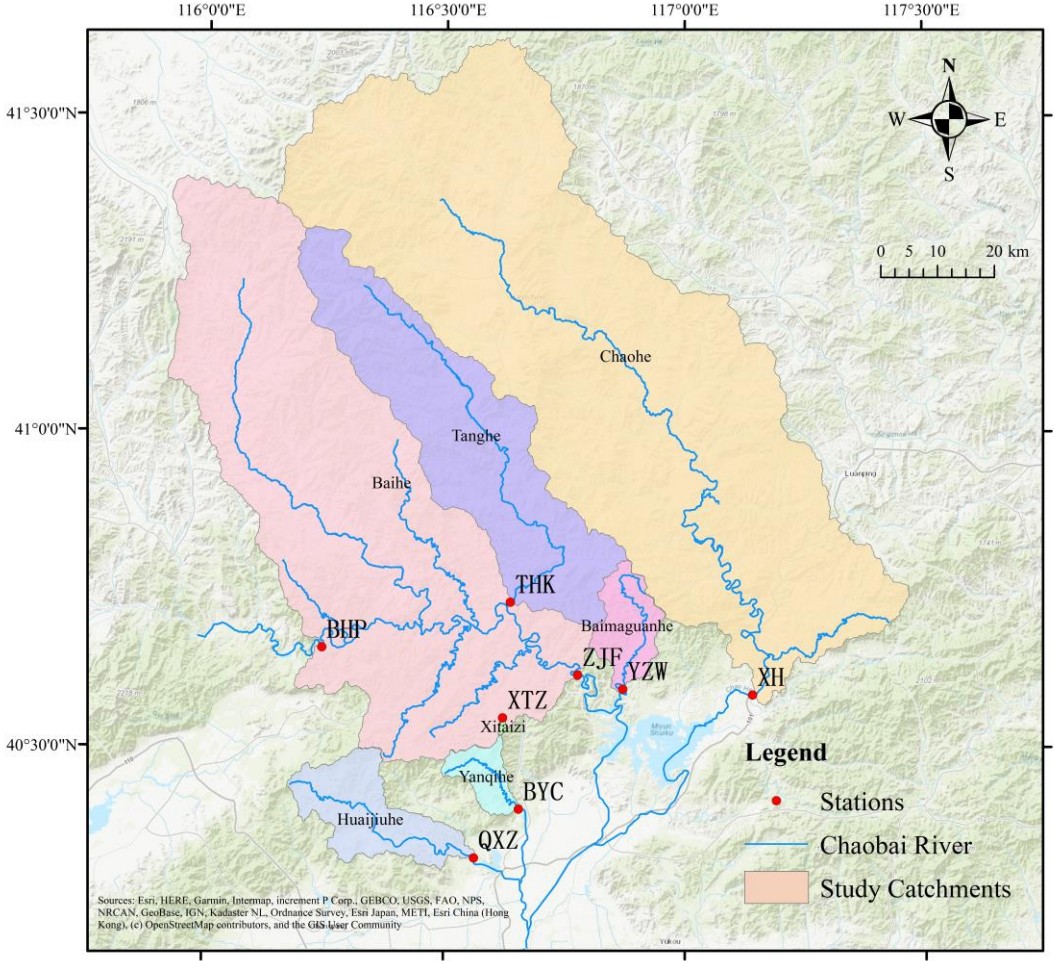

**Figure 1: Geographic distribution of study catchments**

**Table 1. Information of study catchments and data**

| NO. | Basin | Hydrological Station | Abbr. | Drainage area (km²) | Number of rainfall gauges | Rainfall gauging area (km²) |
|---|---|---|---|---|---|---|
| 1 | Xitaizi | Xitaizi | XTZ | 3.11 | 1 | 3.11 |

| 2 | Yanqihe | Baiyachang | BYC | 96.06 | 6 | 16.01 |
| 3 | Baimaguanhe | Yaoziwa | YZW | 180.04 | 8 | 22.51 |
| 4 | Huaijiuhe | Qianxinzhuang | QXZ | 332.85 | 10 | 33.29 |
| 5 | Tanghe | Tanghekou | THK | 1263.13 | 4 | 315.78 |
| 6 | Baihe | Zhangjiafen | ZJF | 4660.91 | 14 | 332.92 |
| 7 | Chaohe | Xiahui | XH | 4845.98 | 13 | 372.77 |

## 2.2 Hydrological model

The study catchments are located in a rocky mountainous region with severe weathering and high vegetation cover (Zheng et al., 2013; Yu et al., 2017). On the basis of intensive hydrological and isotopic observations from the Xitaizi experimental catchment, Zhao et al (2019) found that preferential flow in the heavily weathered granite and shallow soils makes up the majority of the stormflow. Recent studies also indicate that subsurface flow is a significant contributor to flood generation (Addisie et al., 2020; Xiao et al., 2020; Wang et al., 2022). To effectively capture the hydrological processes within the study area, a four-source hydrological model was developed, designed to represent multiple hydrological pathways. The model's structural diagram (Figure 2) illustrates these pathways.

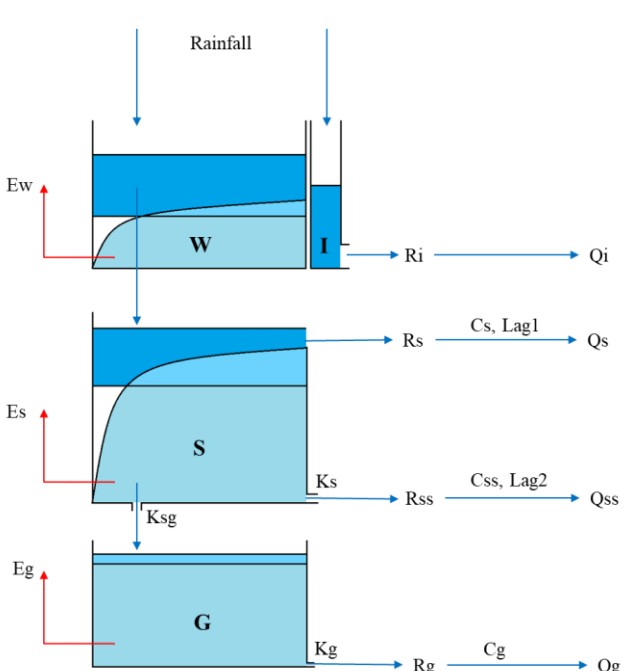

- I,W,S,G: impervious layer, soil layer, subsurface layer, groundwater layer.
- Ri, Rs, Rss, Rg: runoff in impervious layer, surface layer, subsurface layer and groundwater layer.
- Ew, Es, Eg: evaporation in W, S, G layer
- Ks, Kg, Ksg: linear outflow coefficients from S layer, G layer, and from S to G layer
- Cs, Css, Cg: weighting coefficients of Rs, Rss, Rg
- Lag1, Lag2: lag coefficients of Rs, Rss

**Figure 2: The structural diagram of the hydrological model**

The hydrological model is semi-distributed, which first divides the watershed into multiple sub-basins based on the DEM data. Within each sub-basin, the model further divides the surface layer into two representative units in the horizontal direction: pervious and impervious layers. The impervious layer (I Layer) includes waterways, compacted rock layers, and artificial covers (such as concrete roads), among others. Rainfall on the impervious layer is directly converted into runoff impervious layer ($R_i$) for that time step, as follows:

$$R_i = P \tag{1}$$

where $P$ is the precipitation.

The pervious layer is divided vertically into the capillary water layer (W layer), subsurface layer (S layer), and groundwater layer (G layer). To reflect the spatial variability of water storage capacity in the

164 watershed, the W layer and S layer are enclosed by an exponential curve (Zhao, 1992). Rainfall on the

165 pervious layer is partially routed into the W layer, representing soil moisture, which does not contribute

to runoff. Another portion of the rainfall (R) infiltrates into the S layer. Water exceeding the capacity of

the S layer is generated as surface runoff ($R_s$), while the water within the S layer is routed through an

outlet, contributing to subsurface runoff ($R_{ss}$). The equations for surface runoff and subsurface runoff

are as follows:

$$\text{WMM} = \text{WM} * (1+\text{B}) \tag{2}$$

$$A = \text{WMM}\left[1-(1-\frac{W}{WM})^{\frac{1}{1+B}}\right] \tag{3}$$

$$R = P - E_w + W - WM, \quad if\ P - E_w + A \geq WMM \tag{4}$$

$$R = P - E_w + W - WM\left[1 - \left(1 - \frac{P - E_w + A}{WMM}\right)^{1+B}\right], \quad if\ P - E_w + A < WMM \tag{5}$$

$$\text{SMM} = \text{SM} * (1+\text{EX}) \tag{6}$$

$$AU = \text{SMM}\left[1-(1-\frac{S}{SM})^{\frac{1}{1+EX}}\right] \tag{7}$$

$$R_s = R + S - SM, \quad if\ R + AU \geq SMM \tag{8}$$

$$R_s = R + S - SM \times \left[1 - \left(1 - \frac{R + AU}{SMM}\right)^{1+EX}\right], \quad if\ R + AU < SMM \tag{9}$$

$$R_{ss} = S * K_{ss} \tag{10}$$

where WM, SM, B and EX are storage of W, S layer and their exponential coefficients.
Water in the S layer infiltrates into the G layer. The spatial variability of the groundwater layer's storage
capacity is neglected, and groundwater runoff ($R_g$) is calculated using a linear reservoir approach. The
equations for groundwater are as follows:

$$G_t = G_{t-1} + S_t * K_{sg} \tag{11}$$

$$R_g = G * K_g \tag{12}$$

Evaporation occurs in the W, S, and G layers. The evaporation in the W layer is calculated by as follow:

$$E_w = PET * K_{ew} \tag{13}$$

where PET is the mean potential evapotranspiration and $K_{ew}$ is the linear coefficients. $E_s$, $E_g$ are
calculated by similar equations with the linear coefficients of $K_{es}$, $K_{eg}$.
Considering the lag time in runoff response to rainfall, the convergence of surface flow and subsurface
flow on the hillslopes within a sub-basin is modeled using a lag algorithm. No separate lag time is
assigned to groundwater flow, as its runoff response to rainfall is slow, and this behaviour can be captured
through other parameters. No lag time is either assigned to the impermeable surface, as the travel time
of surface water flow within the sub-basin is relatively short and is assumed to not exceed a single time
step. Thus, the equations for the flow from all four pathways are as follows:

$$Q_{i,t} = R_{i,t} * Area * imp/dT \tag{14}$$

$$Q_{s,t} = \left[R_{s,t-1-lag1} * Cs + R_{s,t-lag1} * (1 - Cs)\right] * Area * (1 - imp)/dT \tag{15}$$

$$Q_{ss,t} = \left[R_{ss,t-1-lag2} * Css + R_{ss,t-lag2} * (1 - Css)\right] * Area * (1 - imp)/dT \tag{16}$$

$$Q_{g,t} = \left[R_{g,t-1} * Cg + R_{g,t} * (1 - Cg)\right] * Area * (1 - imp)/dT \tag{17}$$

where $Q_i$, $Q_s$, $Q_{ss}$, $Q_g$ are the flow from the impervious layer, surface layer, subsurface layer and
groundwater layer, $Area$ is the area of the basin, $imp$ is the proportion of impervious area, and dT is
the calculation time step.
The total flow from a sub-basin is the sum of the four flows above. The routing process is modeled using
the Muskingum method (Cunge, 1969; Gill, 1978; Yoon and Padmanabhan, 1993), with the equation
given as:

$$Q_i^t = C_1 Q_{i-1}^{t-1} + C_2 Q_{i-1}^t + C_3 Q_i^{t-1} + (C_1 + C_2)Q_L \tag{18}$$

where i is spatial index, t is temporal index, and $Q_L$ is lateral flow.
In the Muskingum method, the three parameters $C_1$, $C_2$, $C_3$ must satisfy the conditions of being within
the 0-1 range and their sum equaling 1. To accommodate these constraints within the automatic parameter
optimization program, this study reparametrizes the model by optimizing the values of $C_1+C_2$ and $C_1$/
$(C_1+C_2)$, thereby determining the optimal values for the original parameters.

## 2.3 Experimental design for the value of high-resolution data

Daily streamflow and hourly streamflow are important modeling targets in hydrological research and
practice. To test the value of rainfall and measured streamflow data at different resolutions for simulating
streamflow at these two scales, we designed two specific experiments: the daily modeling test and the
hourly modeling test. In this context, 'daily' and 'hourly' refer to the target time scales for the model's
predictions. The flowchart of the tests was shown as Figure 3, and the details are as follows.
(1) Daily modeling test: This test was designed to investigate the impact of high-resolution rainfall data
on daily streamflow simulation. The model was driven by rainfall data at various resolutions (ranging
from 1h to 24h) and calibrated using daily resolution streamflow data. This setup aimed to assess whether
(and to what extent) sub-daily rainfall data can enhance daily streamflow simulation.
(2) Hourly modeling test: This test was designed to investigate the impact of high-resolution input and
streamflow data on hourly streamflow simulation. In this test, the temporal resolutions of input rainfall
data and calibration streamflow data were the same, both set as various resolutions (ranging from 1h to
24h). The model was calibrated using streamflow data with the given temporal resolution, and then the
hourly streamflow simulated by the calibrated model was evaluated based on the hourly measured
streamflow. This setup aimed to determine the necessary data resolution for providing reliable hourly
streamflow simulation.
These experiments aimed to investigate how data resolution affects the accuracy and reliability of
streamflow predictions across various temporal scales. To minimize potential impacts from varying
computational time steps, the hydrological simulations were consistently set to a 1-hour time step for
both tests. This standardization was maintained across all cases, with different input data resolutions used.
Specifically, all input data, including rainfall, were resampled to a 1-hour resolution via prior averaging
before driving the model. As a result, the model's original outputs were always produced at an hourly
scale.
In the daily modeling test, rainfall data at varying temporal scales was input into the hydrological model
to produce simulated hourly streamflow, which was later aggregated to the daily scale for comparison
with observed daily streamflow. Model parameters were then optimized by aligning the simulation with
observations using Python Surrogate Optimization Toolbox (pySOT, Eriksson et al., 2019), aiming to
maximize the Nash-Sutcliffe efficiency (NSE). The optimization process, iterated via Symmetric Latin
Hypercube Design (SLHD), concluded upon convergence or after reaching a 3000-iteration threshold.
After 100 trials, the final parameters were selected based on maximum NSE. Additionally, after
calibration, Relative Error of Peak flow (REP) was computed as a secondary performance metric. These
metrics were calculated as follows:

$$NSE = 1 - \frac{\sum_{t=1}^n \left(Q_t^{obs} - Q_t^{sim}\right)}{\sum_{t=1}^n \left(Q_t^{obs} - \overline{Q^{obs}}\right)} \tag{19}$$

$$REP = \frac{Q_{sim,p} - Q_{obs,p}}{Q_{obs,p}} \tag{20}$$

where, $Q_t^{obs}$ and $Q_t^{sim}$ are the streamflow for the observed and simulated time series, $\overline{Q^{obs}}$ is the average value of the observed streamflow, $Q_{sim,p}$ and $Q_{obs,p}$ are the simulated and observed peak flow, respectively.

The hourly modeling test followed a similar procedure to the daily modeling test, inputting rainfall data at various temporal resolutions into the hydrological model to produce simulated hourly streamflow. This output was aggregated to match the resolution of the input data and compared with the corresponding observed data for calibration. The performance of calibrated model on simulating hourly streamflow was then assessed by calculating NSE and REP, based on the hourly simulated and observed streamflow data. The flowchart of the experimental tests was illustrated in Figure 3, where D and H refer to daily and hourly test, $x_i$ is each member of the time step (t.s.) set (TS), which consists of 1h, 2h, 3h, 4h, 6h, 12h and 24h. $NSE_{D,xi}$ and $REP_{D,xi}$ are the NSE of and REP of daily streamflow forced by rainfall at time step of $x_i$. Similarly, $NSE_{H,xi}$ and $REP_{H,xi}$ denote the NSE and REP for hourly streamflow at time step of $x_i$.

After tests, the paired two-sample t-test, a widely used statistical method to determine whether the means of two related groups of samples are significantly different (e.g., Xu et al., 2017), was adopted to test whether the performance of the hydrological model based on high-resolution data was significantly improved.

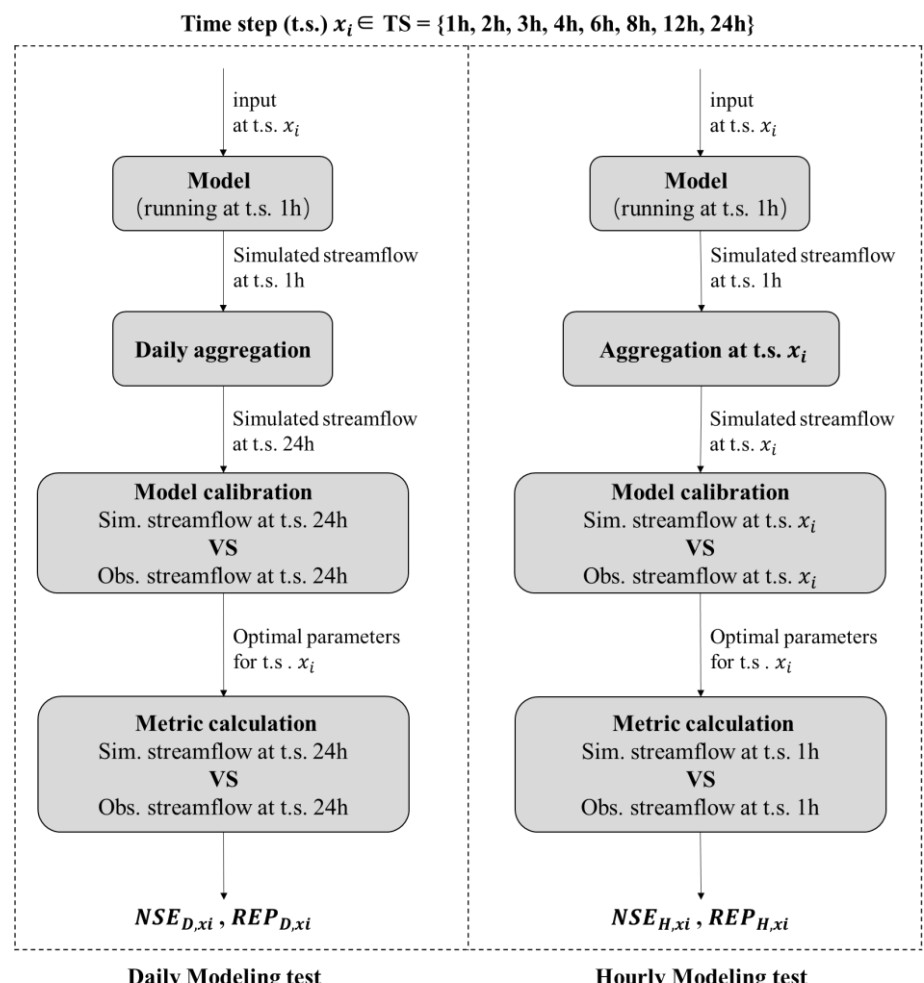

**263**

**264** **Figure 3: Flowchart of the daily modeling and hourly modeling tests**

**265** **2.4 Experimental design for parameters transferability**

**266** To test the potential impact of the resolution of training data and the computational time step on
**267** calibration of model parameters, as well as the transferability of these parameters across different time
**268** scales, we designed two tests: the data resolution test and the computational timestep test. The flowchart
**269** of the tests was shown as Figure 4, and the details are as follows.
**270** (1) Data resolution test: in this test, the model's computational time step was fixed as 1 hour, while the
**271** temporal resolution of the input and measured streamflow varied from 1 hour to 24 hours (as in the hourly
**272** test). Previously, optimal parameter sets, $Par_{xi}$, have been obtained under varying resolutions ($x_i$) of
**273** input and measured streamflow data in hourly modeling tests. In this data resolution test, the optimal
**274** parameter set obtained at one resolution (referred to as the pre-transfer resolution) was used to drive the
**275** model with input data at another resolution (referred to as the post-transfer resolution), resulting in hourly
**276** simulated streamflow. The simulation accuracy, measured by NSE, was then calculated. By comparing
**277** the changes in the simulation metrics obtained by a same set of parameters and different input resolutions,
**278** the transferability of the parameters across varying resolutions was tested.
**279** (2) Computational time step test: in this test, the model's computational time step varied from 1 hour to
**280** 24 hours, while the temporal resolution of the input rainfall and measured streamflow data was fixed as
**281** 24 hours. Firstly, input data at the resolution of 24 hours was fed into the model, and the model was run

at varied time steps, resulting simulated streamflow at varied time steps. Next, the simulated streamflow
was aggregated in daily, and the model parameters were calibrated based on observed daily streamflow.
In this way, the model parameters under different computational steps are obtained. Then, the optimal
parameter set obtained at one computational time step (referred to as the pre-transfer computational time
step) was used to drive the model at another computational time step (referred to as the post-transfer
computational time step), and the NSE was calculated based on the simulated daily streamflow obtained
at this time step. By comparing the changes in simulation metrics, the transferability of parameters
obtained at one computational time step to another was tested.

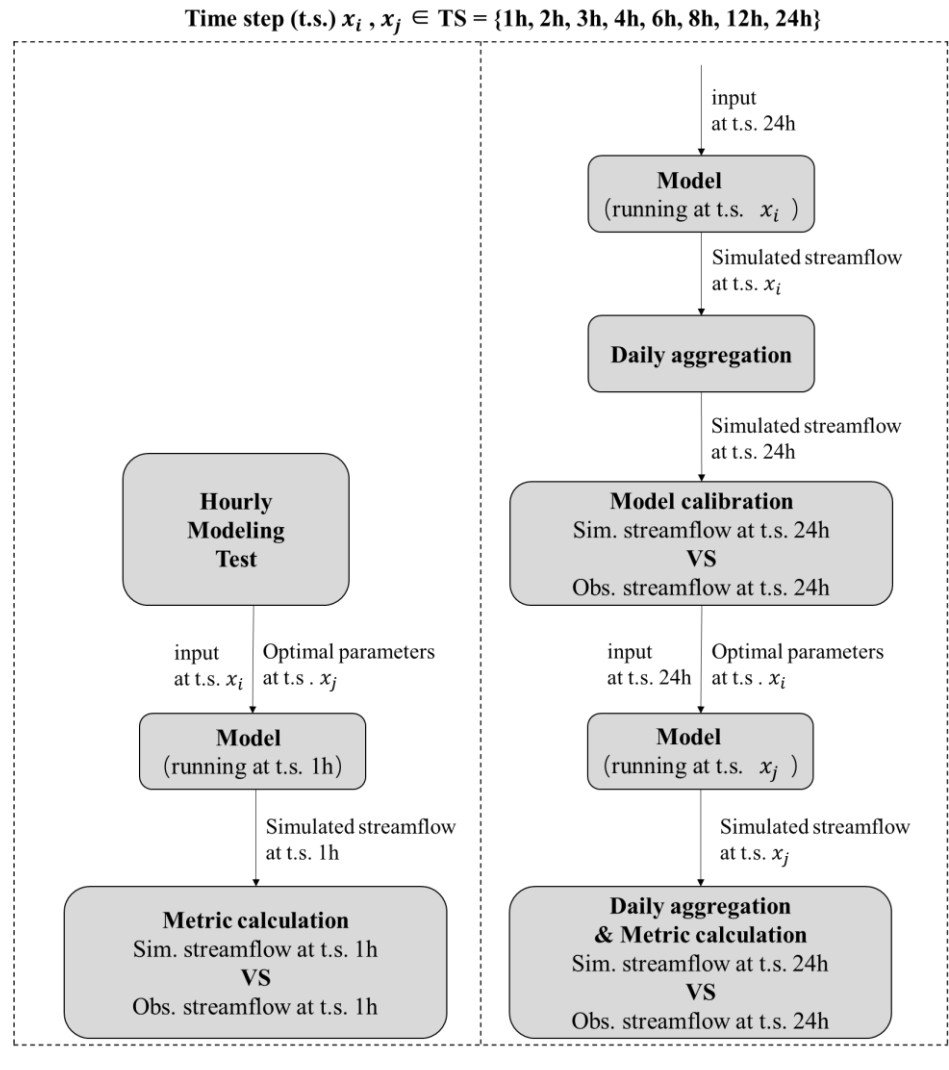

**Figure 4: Flowchart of the data resolution and computational time step tests**
**3 Results**
**3.1 The value of high-resolution data**
The results of the daily and the hourly modeling tests are shown in Figure 5. Subplots (a) and (b) represent
the NSE and absolute values of REP in the daily modeling test, respectively. Subplots (c) and (d) depict
these two metrics in the hourly modeling test. In the daily test, the average NSE obtained by various data
resolutions varied in the range of 0.91 - 0.94. The model performed worst when using 24-hour resolution
data, but even then, the lowest NSE value was 0.82 in the Yanqihe catchment at BYC station, and in the
other 6 catchments, the NSE exceeded 0.89. As for REP, the average $|REP_D|$ at various data resolutions
ranged between 2% and 4% indicating high accuracy in simulation on peak flow at daily scale. In the
hourly modeling test, the metrics got slightly worse compared with the daily test. The average NSE across
various data resolutions ranged from 0.78 to 0.87. The model performed worst when using 24h resolution
data, with the lowest NSE of 0.64, but the NSE exceeds 0.8 in five of the study catchments. The model
produced NSE higher than 0.83 in 6 catchments when using 1h rainfall and streamflow data. The average
$|REP_H|$ varied in the range of 16% - 27%. Compared to the daily modeling test, the model's accuracy in
simulating peak flow declined noticeably in hourly modeling, as the evaluation is more strict. Overall,
these results demonstrated the high performance and reliability of the model in these catchments, with
high NSE and low |REP|.

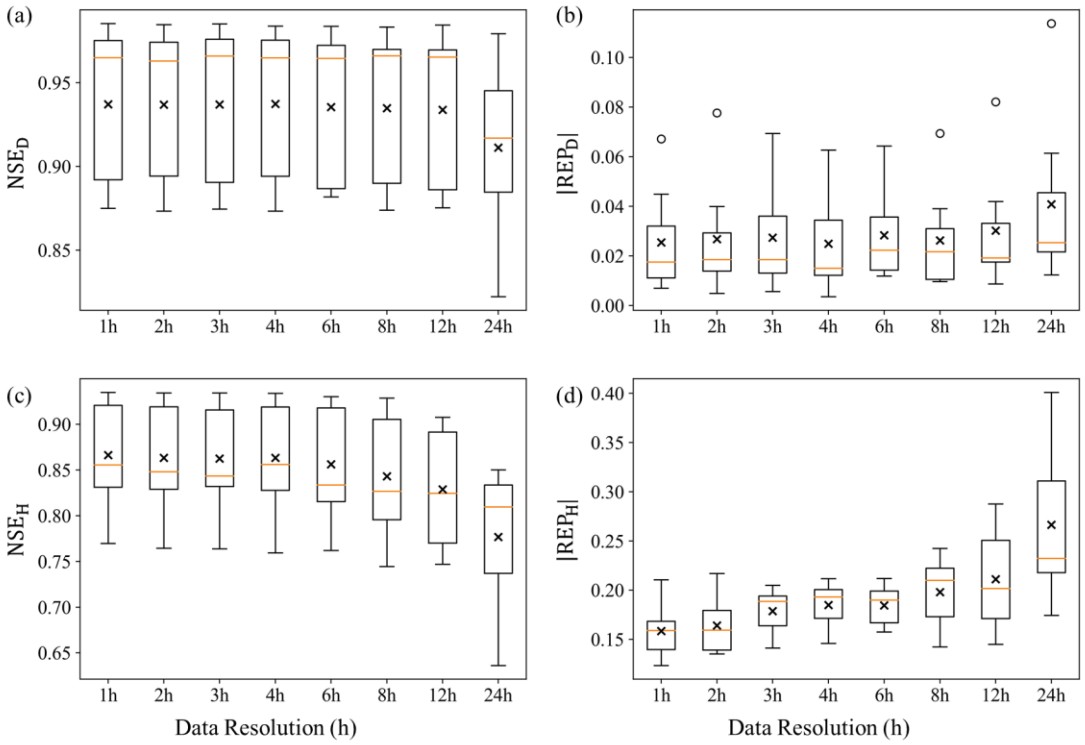


**Figure 5: Box plot of NSE, |REP| in the daily and the hourly modeling tests across 7 catchments**
In both daily and hourly modeling tests, there was an obvious improvement in model performance when
the data resolution increased. For instance, in the daily modeling test, when the data resolution shifted
from 24h to sub-daily 12h, the average NSE increased from 0.91 to 0.93 and the average |REP| decreased
from 4.08% to 3.02%. In the hourly modeling test, the improvement was more obvious. The average
NSE increased from 0.78 to 0.83 and the average |REP| decreased from 27% to 21%, when the data
resolution shifted from 24h to sub-daily 12h. But such improvement got increasingly limited as the
resolution further increased.
To quantify the difference in the model performances when adopting data with different resolutions,
paired two-sample t-tests were conducted, and the results are shown in Table 2. In the daily modeling
test, significant improvement (at 0.05 significance level) on streamflow simulation was brought by sub-
daily (1h – 12h) resolution rainfall data compared to the daily data, as indicated by the low p values in
the last row of Table 2a and Table 2b. However, compared to 12h resolution, only the 1-hour resolution

brought a significant improvement in NSE at the significance level of 0.05. As for |REP|, there were significant differences in |REP| at 2h and 8h resolution compared to 12h resolution. Overall, the results suggested that for daily streamflow forecasting, continuously increasing rainfall data resolution beyond the 12h threshold did not bring significant improvement on model performance. That is, the simulated daily streamflow obtained from a model driven by 12h rainfall input had comparable reliability to that forced by 1h data, and the effect of rainfall data with a temporal resolution exceeding 12h on enhancing daily forecasted flow was negligible.

Similar results were observed in the hourly modeling test (Table 2c and Table 2d). Compared to the daily data, utilizing higher-resolution data effectively enhanced the model's forecasting performance for hourly streamflow. Specifically, regarding the NSE, there were significant differences in the model's performance when using 8h resolution data compared to that obtained by 2h to 6h resolution data. But, when the data resolution reached 6 hours or higher, there were no statistically significant differences in NSEs, indicating that further increasing the resolution did not consistently enhance overall simulation accuracy. Consequently, taking NSE as the performance metric, simulated hourly streamflow obtained by a model driven and calibrated by 6h data was comparably accurate to that obtained by higher resolution data. Data with a resolution higher than 6h did not provide significant additional value. Compared to NSE, the improvement in |REP| was more pronounced with the increase in data resolution in the hourly modeling test. Compared with daily (24h) resolution data, all sub-daily resolution (1h-12h) data showed significant improvement in |REP| (at 0.05 significance level). Comparing the effects of sub-daily scale data, although there was no significant difference in the |REP| when resolutions were close (e.g., 6-hour and 8h resolutions), significant differences in |REP| still existed when the resolution was sufficiently high (e.g., 1h) compared to other resolutions. For instance, the first column of Table 2d indicated that only the |REP| obtained with 2h resolution data showed no statistically significant difference when compared to 1h resolution data. This suggests that continuously increasing data resolution has greater value in improving the accuracy of predictions on peak flow.

**Table 2 P-values of the paired two-sample t-tests for each metric**

**Table 2a P-values of the paired two-sample t-tests for NSE in daily modeling test**

| Resolution | 1h | 2h | 3h | 4h | 6h | 8h | 12h |
|---|---|---|---|---|---|---|---|
| 2h | 0.987 | | | | | | |
| 3h | 0.932 | 0.962 | | | | | |
| 4h | 0.459 | 0.562 | 0.693 | | | | |
| 6h | **0.033*** | 0.175 | **0.043*** | 0.054 | | | |
| 8h | 0.223 | 0.330 | 0.109 | 0.157 | 0.770 | | |
| 12h | **0.041*** | 0.095 | 0.148 | 0.061 | 0.537 | 0.599 | |
| 24h | **0.036*** | **0.042*** | **0.031*** | **0.036*** | **0.039*** | **0.031*** | **0.046*** |

**Table 2b P-values of the paired two-sample t-tests for |REP| in daily modeling test**

| Resolution | 1h | 2h | 3h | 4h | 6h | 8h | 12h |
|---|---|---|---|---|---|---|---|
| 2h | 0.5581 | | | | | | |
| 3h | 0.1446 | 0.8063 | | | | | |
| 4h | 0.6260 | 0.8122 | 0.3503 | | | | |
| 6h | 0.3196 | 0.9739 | 0.7922 | 0.6138 | | | |
| 8h | 0.8420 | 0.6117 | 0.4098 | 0.8532 | 0.3476 | | |

| | 1h | 2h | 3h | 4h | 6h | 8h | 12h |
|---|---|---|---|---|---|---|---|
| 12h | 0.0743 | **0.0164*** | 0.2985 | 0.1927 | 0.2364 | **0.0412*** | |
| 24h | **0.0314*** | **0.0189*** | **0.0490*** | 0.0582 | 0.0763 | **0.0352*** | **0.0497*** |

**Table 2c P-values of the paired two-sample t-tests for NSE in hourly modeling test**

| Resolution | 1h | 2h | 3h | 4h | 6h | 8h | 12h |
|---|---|---|---|---|---|---|---|
| 2h | 0.368 | | | | | | |
| 3h | 0.283 | 0.571 | | | | | |
| 4h | 0.370 | 0.666 | 0.559 | | | | |
| 6h | 0.088 | **0.044*** | 0.109 | 0.096 | | | |
| 8h | **0.037*** | **0.017*** | **0.032*** | **0.028*** | **0.016*** | | |
| 12h | **0.013*** | **0.007*** | **0.010*** | **0.011*** | **0.007*** | **0.028*** | |
| 24h | **0.009**** | **0.007**** | **0.008**** | **0.009**** | **0.008**** | **0.011*** | **0.011*** |

**Table 2d P-values of the paired two-sample t-tests for |REP | in hourly modeling test**

| Resolution | 1h | 2h | 3h | 4h | 6h | 8h | 12h |
|---|---|---|---|---|---|---|---|
| 2h | 0.327 | | | | | | |
| 3h | **0.006**** | 0.084 | | | | | |
| 4h | **0.001**** | **0.009*** | 0.194 | | | | |
| 6h | **0.000**** | **0.001**** | 0.113 | 0.378 | | | |
| 8h | **0.005**** | **0.006*** | 0.145 | 0.123 | 0.411 | | |
| 12h | **0.018*** | **0.023*** | 0.066 | 0.066 | 0.149 | 0.112 | |
| 24h | **0.011*** | **0.015*** | **0.018*** | **0.020*** | **0.036*** | **0.031*** | **0.016*** |

Note: ** and * indicates significance at 0.01 and 0.05

## 3.2 Parameters transferability across data resolutions

The optimized model parameters at various data resolutions were obtained under a fixed computational time step of 1-hour in the hourly modeling test. To assess the transferability of these parameters under different data resolutions, the data resolution test was conducted following the experimental design outlined in Section 2.4. The results are shown in Figure 6. In each subplot, each curve represents the NSE values obtained when the optimal parameters calibrated from a specific input resolution are transferred (without any transformation) to drive the model with other input resolutions.

First, when examining the differences among the curves, it was found that in most catchments, the curve representing the 24h resolution consistently fell below the others. This aligns with the results from the previous section, indicating that the model's performance was the lowest when using 24h resolution rainfall and streamflow data. When these parameters are transferred to other resolutions, they also exhibited the lowest performance.

In all catchments except for XTZ, when parameters calibrated with a specific data resolution were transferred to other resolutions, simulation accuracy improved as the resolution of the data used increased. Notably, when the resolution increased from 24h to 12h, the NSE showed the most significant improvement. However, when the input data resolution ranged between 1h and 8h, the NSE remained relatively stable. This observation is consistent with the results and conclusions from Section 3.1. Even though there were some variations in model performance when parameters were transferred to other time scales, the performance remained acceptable, with the lowest NSE still exceeding 0.5. This lowest NSE occurred at the QXZ station when the pre-transfer resolution is 6h and post- transfer resolution is 24h.

When the post- transfer resolution was finer than 24h, the NSE at QXZ was consistently above 0.7. Overall, after parameter transfer, the model continues to demonstrate satisfactory simulation performance.

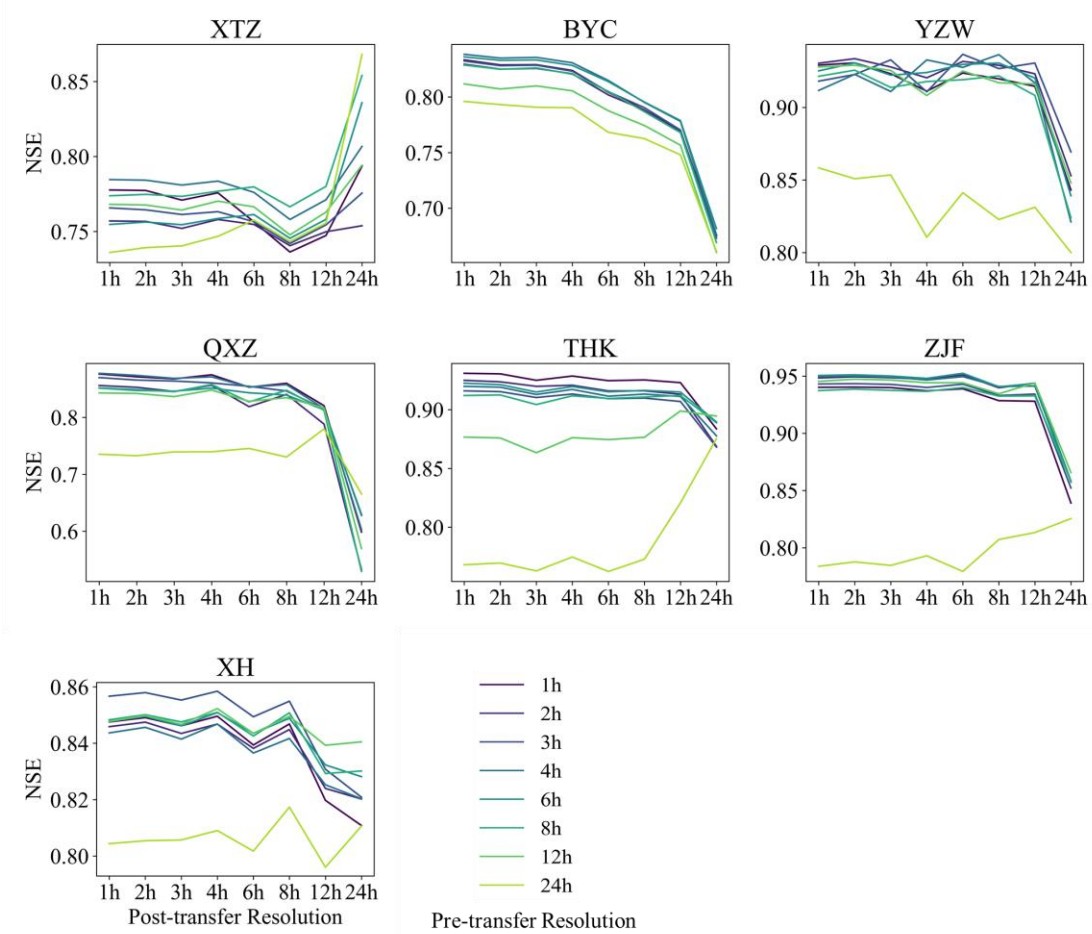

**Figure 6: The NSE values after transferring the parameters obtained at one resolution to other resolutions**

**3.3 Parameters transferability across computational time steps**

To assess the transferability of parameters under different computational time steps, the computational time step test was conducted following the experimental design outlined in Section 2.4. The results are shown in Figure 7. The value in the row i and column j represents the NSE value obtained when transferring the parameters calibrated with a computation time step of $x_i$ directly to a model with a computation time step of $x_j$ ($x_i$, $x_j$ ∈{1h, 2h, 3h, 4h, 6h, 8h, 12h, 24h}, referred to as pre-transfer and post-transfer computational time step, respectively). The values on the diagonal represent the NSE values obtained when running the model with a specific computational time step and calibrating the parameters with daily streamflow. In this case, the parameters were not transferred (i.e., the pre-transfer and post-transfer time steps are the same). First, the values on the diagonal are all greater than 0.7, with most exceeding 0.85, and the average is 0.88. This indicates that the model performs well across different computation time steps, further confirming its reliability. Secondly, within each basin, the values on the diagonal are very close to each other, implying that when both the input rainfall data resolution and the output streamflow resolution are at the daily scale, nearly identical simulation accuracy can be achieved regardless of the computation time step used (within the 1h-24h range).

When parameters calibrated at one computation time step were transferred to other computation time

steps (values in the same row in the Figure 7), the NSE values varied significantly. Compared to the results with the data resolution test in Section 3.2, the variation in NSE under the varying computation time step was much greater. In many cases, the NSE value after transferring parameters was even less than 0, indicating that the model parameters lose their transferability (with unreliable accuracy) when the model's computation time step is varied. Notably, in each subfigure, the values in the lower left part are even lower than those in the upper right part, suggesting that the model's performance is particularly unreliable when parameters calibrated at larger computation time steps are transferred to smaller ones.

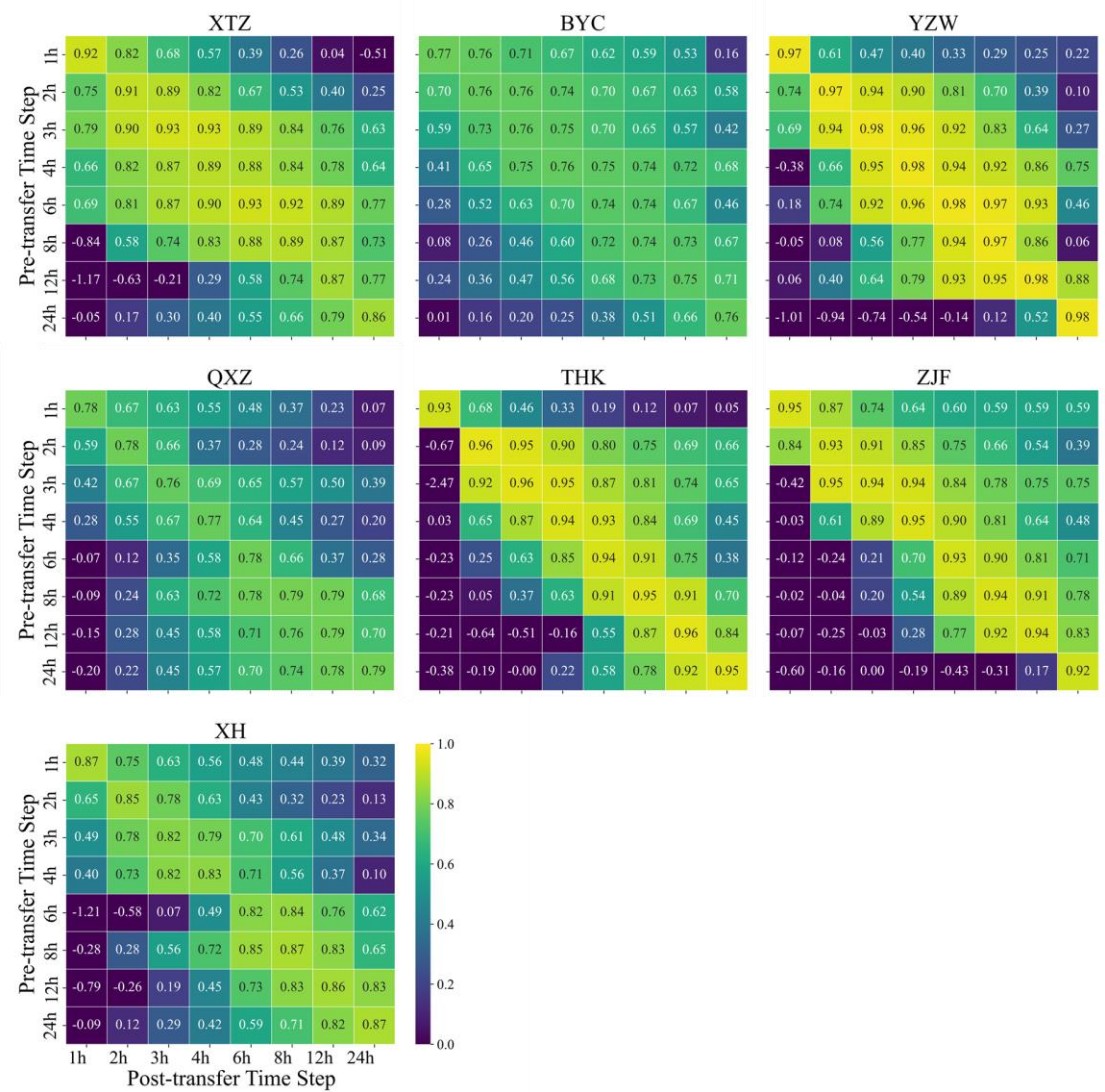

**Figure 7: NSE values after transferring the parameters obtained at one computation time step to other time steps.**

## 4 Discussion

### 4.1 Potential factors for the limited impact of high-resolution data

The results indicated that increasing input data resolution, especially from 24 to 12 hours, significantly boosted simulation accuracy for daily streamflow, consistent with expectations regarding the benefits of

high-resolution data. However, beyond the 12-hour mark, performance became marginal or even declined. Similar patterns emerged in hourly simulations, where benefits of finer-than-6-hour data were negligible or negative, contradicting the intuitive expectations that higher-resolution data always enhances hydrological models. Similar findings were reported by previous studies that investigated the effects of temporal resolution on hydrological models across different regions and model types. Ficchì et al. (2016) explored 240 catchments in France using the GR4 rainfall-runoff model across eight temporal scales, ranging from 6 minutes to 1 day. Their analysis revealed that, on average, finer resolution data provided no additional value when model outputs were aggregated to a 6-hour reference scale. Similarly, Reynolds et al. (2017), while calibrating the HBV model in two small Central American basins, observed that using daily streamflow data produced results comparable to those obtained with sub-daily resolution.

Another notable result we observed is that in the Hourly Test, when the resolution reached or exceeded 6 hours, there was no significant improvement in the NSE, while the |REP| ceased to improve significantly only the resolution reached 2 hours. In our previous study in southern China, both of these threshold resolutions were found to be 6 hours. On one hand, this indicates that the threshold resolution for limiting further improvements in model performance depends on the evaluation metrics used, as each metric emphasizes different aspects of the time series and comes with its own limitations and trade-offs (e.g., Schaefli and Gupta, 2007; McMillan et al., 2017; Fenicia et al., 2018). The benefits of high-resolution data may not be fully captured by a single measure, and using different metrics to assess the impact of high-resolution data on model performance may lead to varying conclusions. On the other hand, a deeper investigation into the causes may point to differences in the climate and runoff characteristics of the two study regions. Southern China features subtropical and tropical monsoon climates, with warm, humid conditions and abundant, evenly distributed rainfall (Fan et al., 2019; Domrös and Peng, 2012). Annual precipitation typically exceeds 800 mm (averaging 1500 mm in our previous study area), classifying it as a humid region. Flood generation in this kind of humid region is predominantly governed by saturation excess (Dunne et al., 1975; Zhao, 1992; Manfreda, 2008). In contrast, northern China experiences a temperate monsoon climate with lower and more concentrated rainfall. Annual precipitation is generally below 800 mm (averaging 600 mm in this study area), making it a semi-humid to semi-arid region where flood generation is primarily driven by infiltration excess and subsurface preferential flow (Zhao, 1992; Zhao et al., 2019; Fu et al., 2024). Hence, in arid catchments, the enhancement of temporal resolution of data is more conducive to improving the model's performance in simulating peak flow as compared to humid catchments, as high temporal resolution data enables more precise capture of variations in precipitation and discharge, particularly at peak values.

While the catchments and models vary across different studies, the overall findings are largely consistent, suggesting that simply increasing data resolution doesn't always lead to better model performance. Several factors may limit the additional benefits of higher resolution data. Firstly, a straightforward reason could be the choice of the evaluation metric. In the hourly modeling test, when the resolution exceeded 6 hours, there was no significant improvement in the NSE, but the |REP| showed a marked change. In some cases, different metrics may conflict with each other, making it impossible to optimize them simultaneously. The benefits of high-resolution data to a model might not be fully captured by the commonly used metrics and may require alternative indicators for quantification. Using different metrics to assess the impact of high-resolution data on the model could lead to varying conclusions, highlighting the complexity and multifaceted nature of the benefits conferred by enhanced data granularity. Secondly, due to spatial and temporal autocorrelation in variables like rainfall and runoff, increasing resolution

beyond a certain threshold may not provide effective new information. There may be no significant difference between actual high-resolution data and high-resolution data obtained by resampling from coarser data. The extent of this difference is related to the characteristics of the climate of the catchment and its runoff generation processes. Thirdly, model input data, particularly rainfall, may have a lower signal-to-noise ratio at higher temporal resolutions due to difficulties in data validation and increased uncertainty in areal average rainfall estimates (Ficchì et al., 2016; Moulin et al., 2009). Besides, since hydrological models inherently simplify natural processes, they may dampen the natural smoothing effect seen in rainfall-runoff interactions. As a result, using high-resolution temporal data to drive the model could introduce excessive variability in the simulated flow, potentially degrading the model's performance. Finally, the model's structure might not be adequately designed to handle the added complexity that comes with shorter time steps. Melsen et al. (2016) pointed out that calibration and validation time intervals should align with the spatial resolution to accurately capture the relevant processes. Some empirical formulas within the model may not be applicable at shorter time scales.

## 4.2 Parameters transferability across data resolutions

The results in Section 3.2 indicated that when the computation time step is fixed at 1-hour, the model demonstrated good performance even when parameters are transferred to input conditions with different resolutions. As shown in Figure 6, in most cases, as the input resolution improved, the NSE also increased. However, some exceptions were found. At hydrological stations such as the THK and ZJF, when using parameters calibrated with 24-hour data, there was an increase in NSE as the rainfall resolution decreased. At the XTZ station, NSE also increased when the rainfall resolution dropped below 8 hours, regardless of the parameters used. This anomaly was particularly pronounced at the THK station. Conversely, at the BYC station, the NSE consistently decreased as the rainfall resolution decreased across all parameters. We selected the THK and BYC stations as representative cases and compared the streamflow processes driven by 1h and 24h rainfall resolutions using parameters calibrated with 24h data (as shown in Figure 8). Based on these flow processes, we explored the reasons behind these observed phenomena.

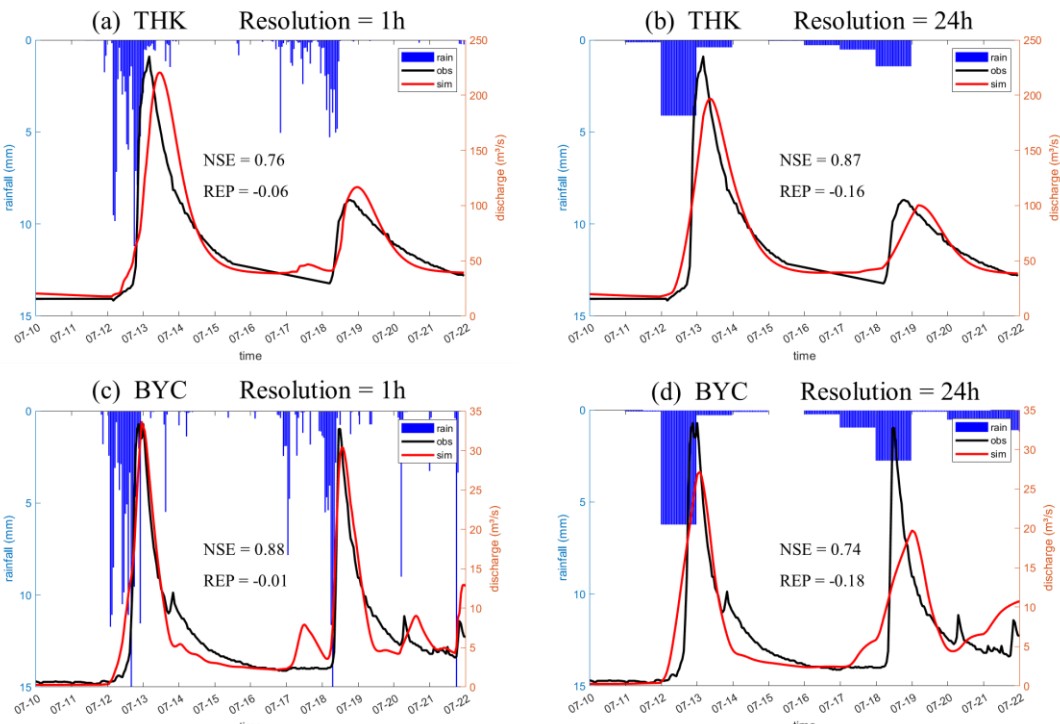

**Figure 8:  streamflow processes at THK and BYC driven by 1h and 24h rainfall resolutions using parameters calibrated by 24h data**

In Figures 8(a) and (b), the model parameters were calibrated using 24h data, but the rainfall data used to drive the model were at resolutions of 1h and 24h, respectively. The same setup was applied in Figures 8(c) and (d). We observed that when using 1h resolution rainfall data, the simulated value of the first flood peak at the THK station was closer to the measured value, even though the NSE at 1h resolution was statistically lower than the NSE at 24h resolution.

To more comprehensively evaluate the simulation accuracy and the impact of different parameters, we conducted further analysis. As mentioned in Section 2.2, we ran 100 iterations using the pySOT program for parameter calibration, which resulted in 100 sets of optimized parameters. Using these 100 parameter sets and the rainfall data at both 1h and 24h resolutions, we evaluated the simulation accuracy of the THK station's streamflow using NSE, KGE, and REP indicators, as shown in Figure 9.

Among the results obtained using the 100 sets of optimal parameters, the NSE values driven by 1h resolution rainfall data were generally lower than those driven by 24h resolution rainfall, with average values of 0.63 and 0.77, respectively. The KGE values were relatively close under both resolutions, with average values of 0.81 and 0.84, respectively. As for the |REP| indicator, the trend was reversed, with the 1h resolution rainfall data yielding better results than the 24h resolution data, with average |REP| values of 9% and 16%, respectively. Based on the runoff processes shown in Figure 8 and the different indicators in Figure 9, we infer that the observed phenomenon, where simulation accuracy decreases as resolution increases, may be related to the evaluation metrics used and the flood characteristics of the basin.

Compared to the BYC station, the THK station exhibited a slower streamflow process during flood events, particularly during the recession phase. We defined a concept similar to half-life period, denoted as $T_{hl}$, to characterize the rate of flood recession. $T_{hl}$ is the time taken for the streamflow to decay from its peak to half of the peak value. At the THK station, $T_{hl}$ is 16 hours, while at the BYC station, $T_{hl}$ is 8 hours, indicating that the flood recession at THK is slower than at BYC. In catchments with a more gradual recession, observed streamflow at a 24h resolution does not provide as much effective information for

model's calibration as higher-resolution data. Furthermore, when 24h resolution rainfall is used as input and 1h as the computational time step, the model tends to produce a smoother simulated streamflow process, since it distributes the rainfall evenly over each hour. Consequently, parameters related to flow routing are not accurately calibrated. As a result, when the model is driven by higher resolution rainfall data such as 1h, larger errors occur in the predicted peak time. However, when using 24h resolution rainfall data, the smoothing effect of the 1h computational time step leads to a simulated recession process that more closely matches the observed values, thus improving the NSE.

**Figure 9: Metrics at THK station using 100 sets of parameters and different resolutions of rainfall**

The results indicated that when the computational time step is fixed as 1h, parameters calibrated under different data resolutions can be transferred and used in models with other resolutions. To further explain the transferability of parameters and identify any patterns as resolution changes, we compared parameters across different resolutions. However, due to the parameter equifinality (Her and Chaubey, 2015; Foulon and Rousseau, 2018), a single optimal parameter set may not be representative enough to accurately reflect the patterns. Therefore, we analyzed 100 sets of parameters calibrated at each resolution, with partial results shown in Figures 10-12. The findings revealed that most parameters did not exhibit a significant and consistent trend of variation with changes in resolution. In other words, parameters calibrated under different resolutions showed little variability, which explains their transferability across resolutions. However, some parameters did show a certain consistent trend with resolution changes. Figure 10 illustrates the trend of the parameter Lag1 with changes in resolution. This parameter in the model reflects the lag time of surface runoff (the time from the generation of surface runoff until it reaches the outlet of the sub-basin). As the resolution becomes coarser (from 1h to 24h), the effective information provided by the observed streamflow to the model decreases, and the requirement for precision in peak time also reduces in the calibration period. This relaxation in constraints led to an increase in both the mean value and the range of variation of Lag1. Notably, at stations XTZ, THK, and ZJF, when the data resolution is 24h, the mean value of Lag1 exceeds 10h or even 15h, showing a significant difference from the value at 1h resolution. The substantial divergence in the optimal parameters for models employing low-resolution versus high-resolution data results in a decline in model

efficacy. This decline occurs even when high-resolution data is employed, provided that the parameters are those optimized for low-resolution data scenarios. In contrast, at stations BYC, YZW, and QXZ, when the data resolution is 24h, the mean value of Lag1 is less than 5h, which is not significantly different from the value at 1h resolution. This provides an explanation for why the NSE at XTZ, THK, and ZJF stations exhibits a decline when using high-resolution data, whereas other stations do not experience such a decrease.

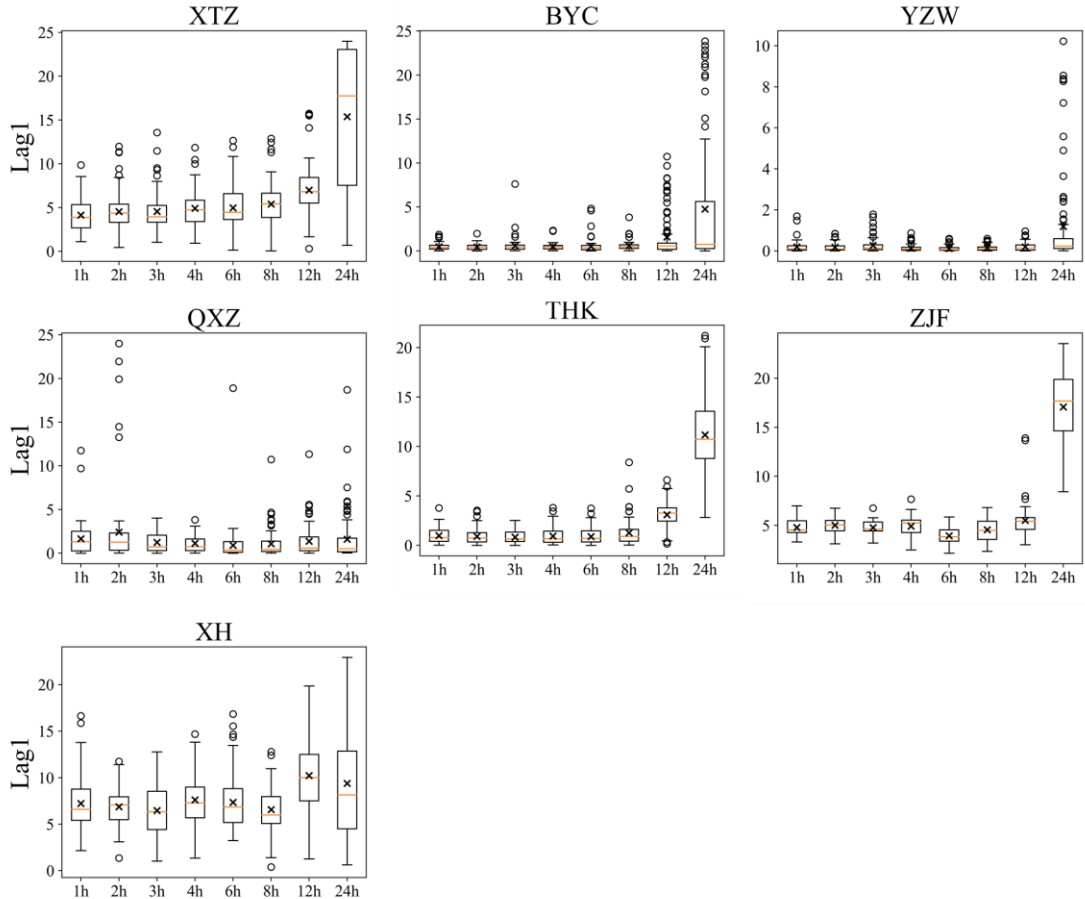

**Figure 10: optimized values of Lag1 across various resolutions**

The parameter $C_1+C_2$ also exhibited a regular trend of variation with changes in resolution (Figure 11). Generally, the larger this parameter, the faster the model's runoff responds to rainfall, resulting in a flood process that rises and falls sharply. When the time resolution is coarse, the variability of runoff may not be fully captured in the observed data. As a result, a model calibrated by a coarser resolution data tend to produce a smoother streamflow process. This is evident at stations such as YZW and QXZ, where the optimized $C_1+C_2$ value decreased as the resolution became coarser. However, we also observed that at most stations, including XTZ, THK, ZJF and XH, this parameter increased as the resolution became coarser. This may be due to the model's computational time step of 1h; when driven by coarse-resolution data, the input data are averaged over each hour, causing the runoff to be smoothed. Consequently, a larger $C_1+C_2$ value was selected by the parameter optimization algorithm to counterbalance this excessive smoothing.

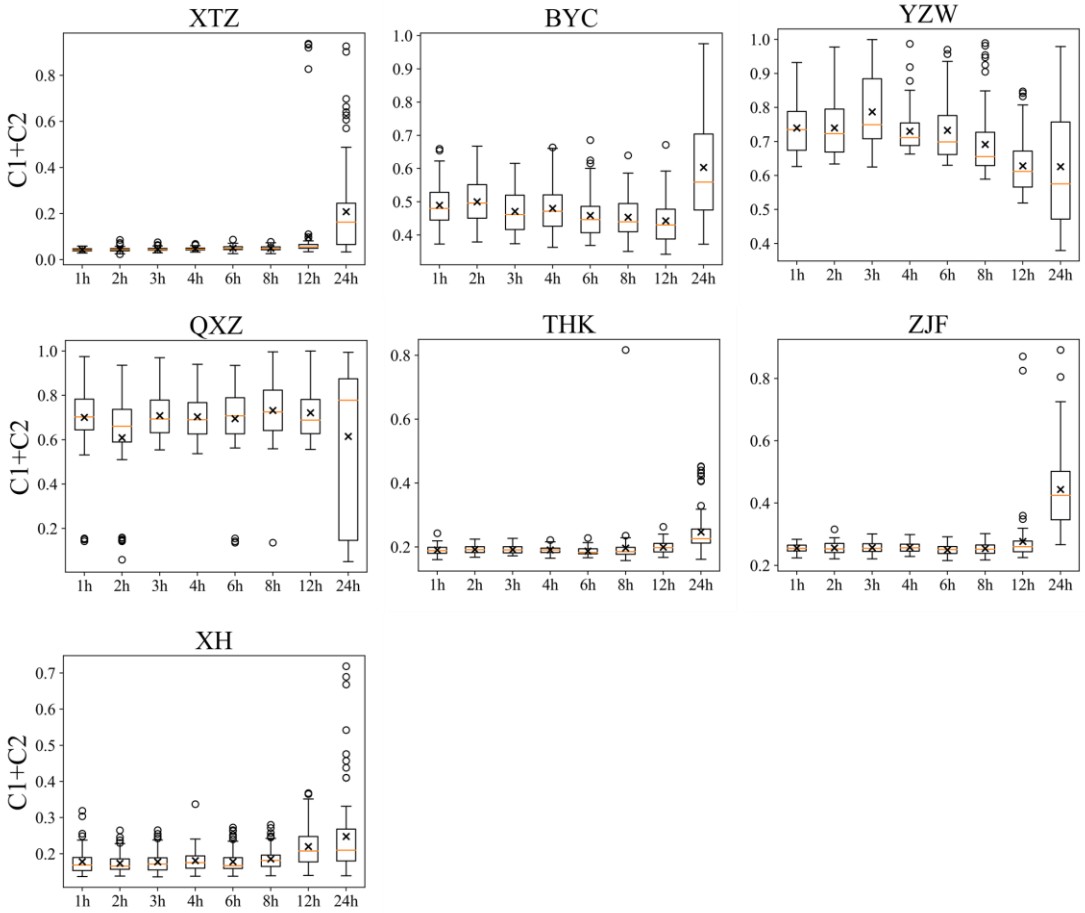


**Figure 11: optimized values of $C_1+C_2$ across various resolutions**

Besides, in certain catchments, specific parameters exhibited regular changes across varying resolutions.
At BYC station, the parameter Ksg decreased as the resolution became coarser. Ksg represents the ratio
of water transfer from the shallow subsurface layer to the deep groundwater layer. A decrease in Ksg
would lead to the shallow subsurface layer becoming saturated more easily, resulting in more surface
runoff. Similarly, at YZW station, the parameter Kg decreased with coarser resolution. Kg represents the
ratio of water conversion from the groundwater layer to groundwater runoff. A reduction in Kg would
cause the groundwater layer to saturate more readily, also indirectly leading to increased surface runoff.
The 1h computational time step evenly distribute rainfall under coarse resolution, which reduces the
simulated peak runoff compared to the actual peak. Therefore, the lower Ksg and Kg values improve
simulation accuracy under coarse resolution conditions by increasing surface runoff.

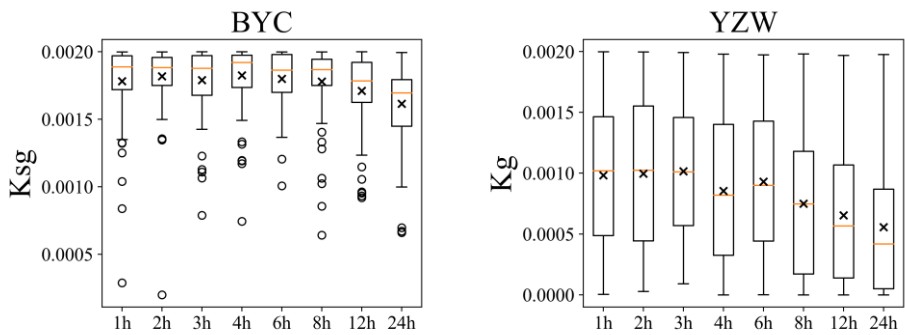


**Figure 12: optimized values of Ksg at BYC station and Kg in YZW station across various resolutions**

**4.3 Parameters transferability across computational time steps**

In contrast to the results of data resolution test, the findings from computational time step test indicate that, model parameters are not transferable across different computational time steps. To explain this non-transferability and identify the primary parameter responsible, we conducted the following test: each parameter from the optimal parameter set corresponding to ranging from 2 to 24 hours time step was sequentially replaced with the optimal parameters for the time step of 1 hour, and the change in NSE coefficient was observed when transferring only one parameter. Depending on the catchment and time step variations, the sensitive parameters causing significant changes in NSE differed; however, a common pattern emerged: in all catchments, two Muskingum parameters were consistently sensitive, and their influence varied systematically with the time step (as shown in Figure 13). In the figure, ΔNSE represents the difference in NSE between using the optimal parameters for the respective time step and those transferred from the 1-hour time step. As the distance between the source and destination time steps increases, the change in NSE caused by parameter transfer becomes more significant. In the XTZ catchment, the NSE is influenced by $C_1 + C_2$ rather than by $C_1 / (C_1 + C_2)$, because this catchment is very small and has not been subdivided into sub-catchments. It consists of only a single hydrological response unit, where the inflow in the routing process is solely lateral flow. By combining Figures 7 and 13, it can be concluded that these two Muskingum parameters have a significant impact on model performance and are one of the main reasons for the non-transferability of model parameters across different computational time steps.

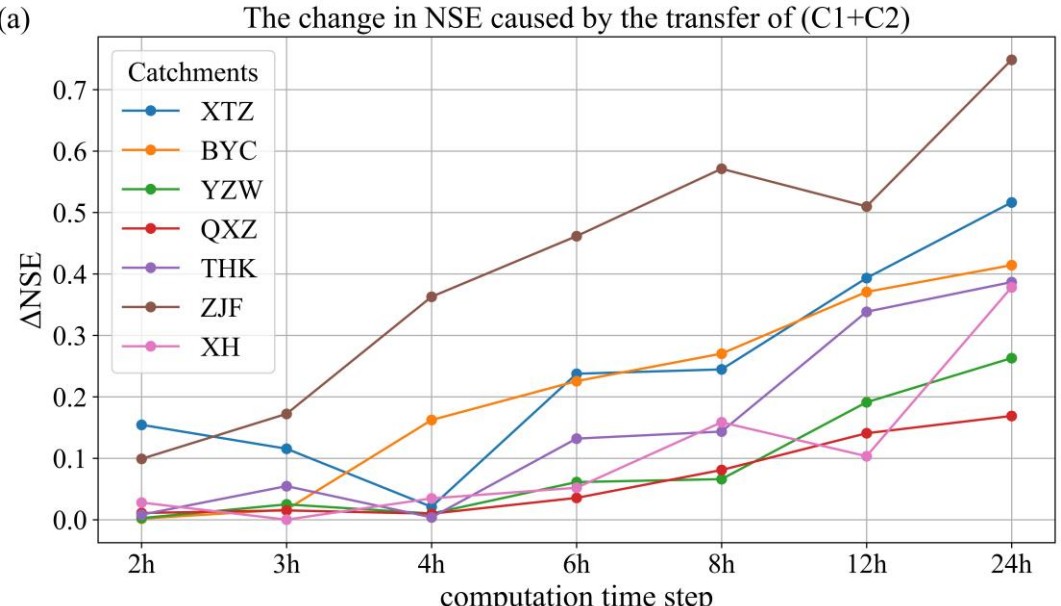

(a)

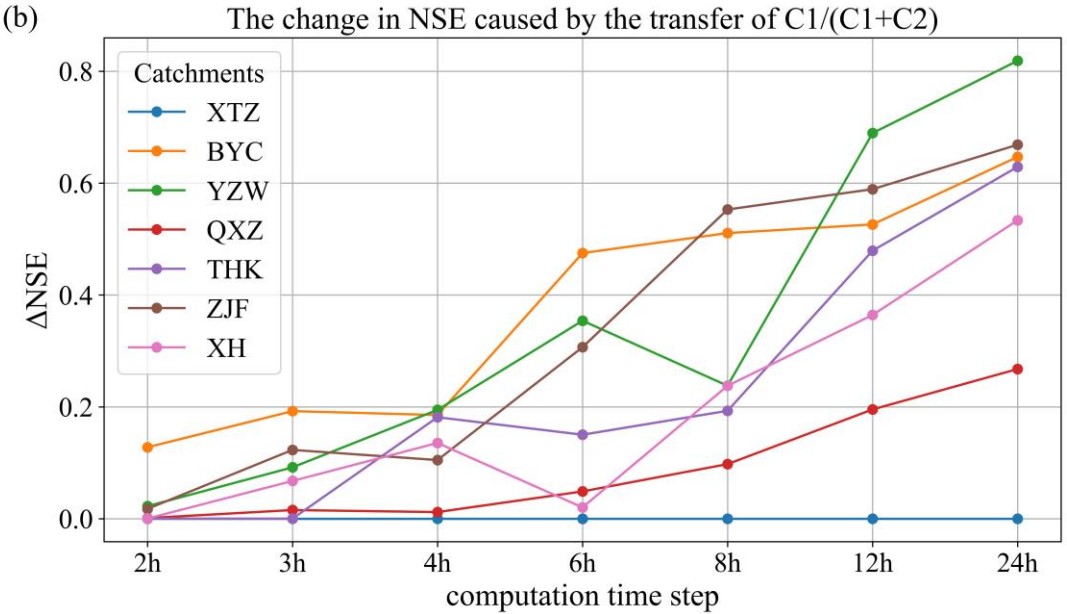

**Figure 13 The change in NSE caused by the transfer of: (a) $C_1 + C_2$; (b) $C_1/(C_1+C_2)$**

To further investigate the variation patterns of parameters with respect to model's computation time step, we analyzed the distribution characteristics of the top 20 best-performing parameter sets out of 100 optimal parameter sets. Unsurprisingly, the two Muskingum-related parameters exhibited clear variations with the change in model's computation time step (as shown in Figure 14 and 15). Specifically, $C_1 + C_2$ increased as the time step grew (indicating that $C_3$ decreased, since the sum of the three parameters equals 1), suggesting that the weight of the inflow term in the routing process increases with the time step, while the correlation between the outlet flow at the end of the current time step ($Q_i^t$) and that at the end of the previous time step ($Q_i^{t-1}$) weakens. The value of $C_1 / (C_1 + C_2)$ decreased as the time step increased, implying that $C_2$ increased. This indicates that as the time step lengthens, the inflow from the upstream ($Q_{i-1}^t$) and the runoff generated within the current time step ($Q_L$) is more likely to be routed to the catchment outlet, enhancing the correlation between the outlet flow at the end of the current time step and the runoff produced within that time step. Furthermore, the variability of $C_1 + C_2$ was notably greater than its variability across different data resolutions (as shown in Figure 11), further confirming that its variability is one of the reasons for the non-transferability of parameters across different time steps. Figure 14 and 15 illustrate the variation patterns of the optimal Muskingum parameters for different computation time steps, which could serve as a reference for researchers and practitioners in the field of hydrological modeling.

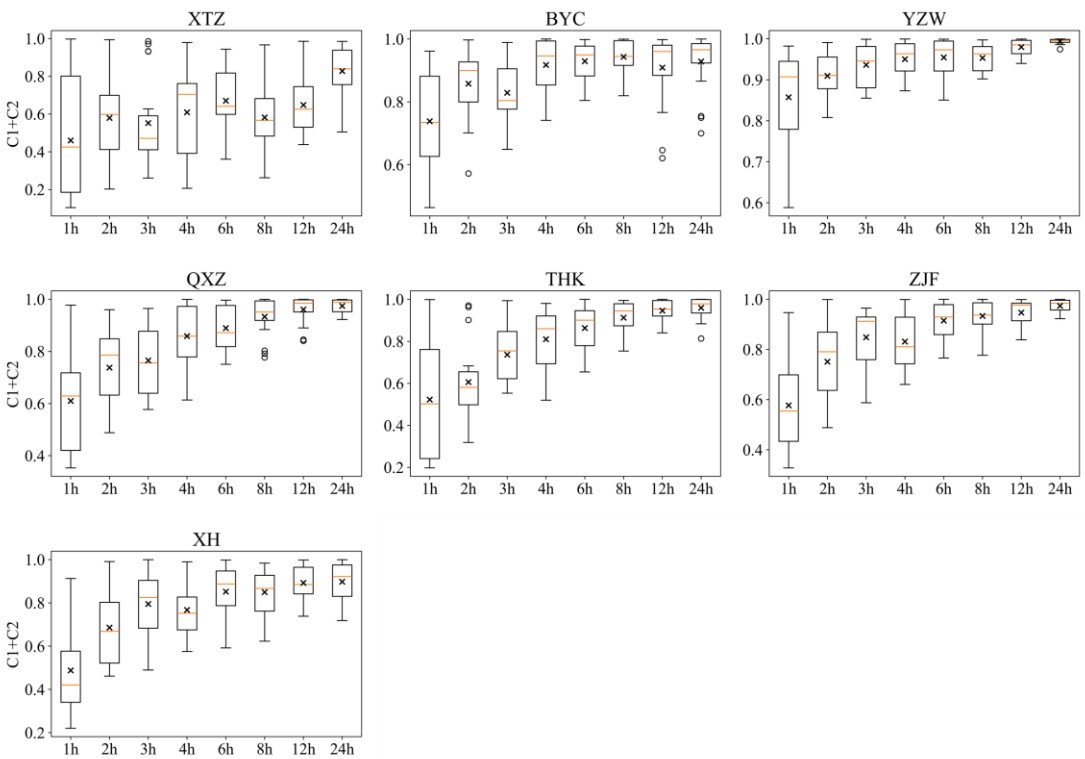

**Figure 14 optimized values of $C_1+C_2$ across various computation time steps**

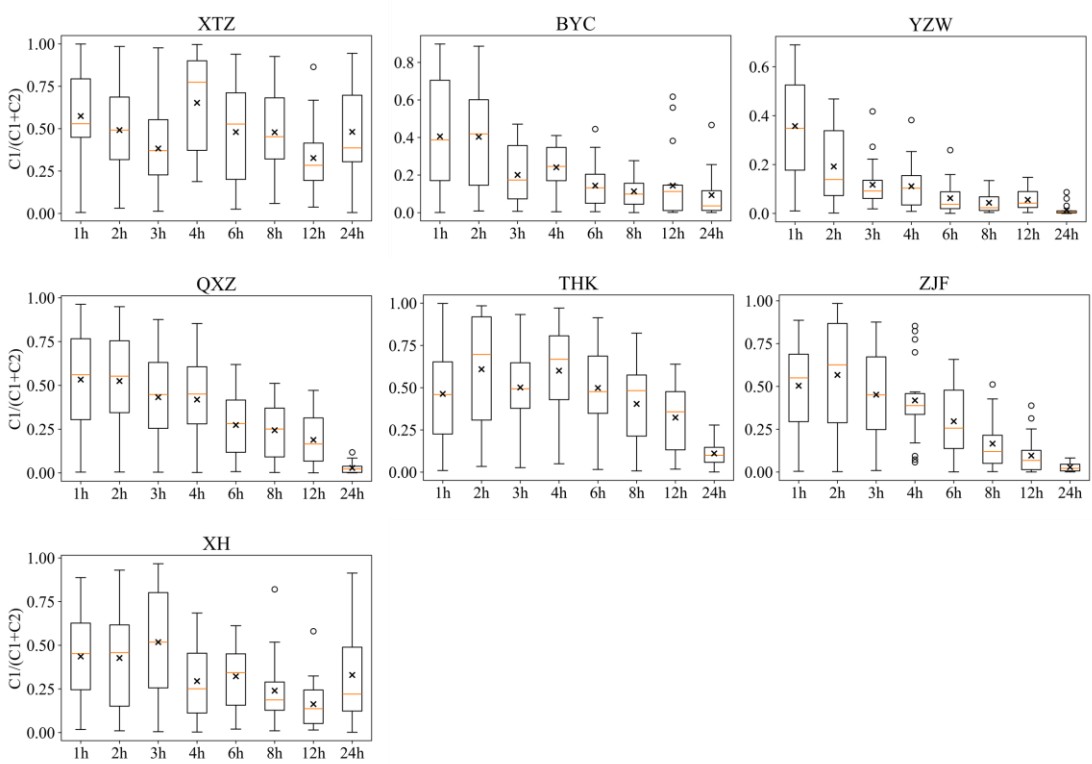

**Figure 15 optimized values of $C_1/(C_1+C_2)$ across various computation time steps**
**4.4 Implications for the selection of data resolution and computation time step**
The findings of this study offer several key insights for building hydrological models with limited data.
1) Data Resolution Considerations:

For daily runoff simulations, it is found that a data resolution of 12h is sufficient to provide accurate simulation results with relatively high precision. This suggests that higher resolution data may not yield significant additional benefits for daily scale modeling. However, for hourly runoff simulations, the adequacy of data resolution depends on the specific objectives of the simulation. If the primary focus is on capturing the overall flood process, such as total runoff volume and approximate duration, a 6h resolution is adequate. On the other hand, if the simulation aims to achieve higher accuracy in peak flow estimation, employing data with finer temporal resolution can enhance the precision of these predictions. This offers practical insights for building numerical models and establishing monitoring stations, suggesting that high-resolution monitoring may not always be necessary. It is essential to balance the additional information gained from higher resolution against the associated costs, aligning with our objectives, enabling efficient resource allocation and ensuring that expenditures yield valuable returns.

2) Selection of Computational Time Step:

Regardless of whether the model is intended for daily or hourly runoff simulations, and irrespective of the input data resolution, it is advisable to adopt a smaller computational time step when constructing the model. This is because the results showed that the simulation accuracy on the coarse scale (24h) with different computation time steps is almost the same, while the model running at a smaller computation step can produce results on a finer scale, which provides the possibility for further analysis. And the model's performance is particularly unreliable when parameters calibrated at larger computation time steps are transferred to smaller ones. This approach also ensures that the model parameters remain applicable across different data resolutions, thereby enhancing the model's flexibility and enabling it to generate accurate simulation results across a range of temporal scales. With the appropriate spatial scale and sufficient computational capacity, opting for a lower computational time step can make the model better equipped to maintain robust performance under varying input conditions and produce results at more time scales, which is crucial for ensuring the transferability of the model parameters and achieving consistent results.

**5 Conclusions**

**5.1 Summary**

This study assessed the value of different resolution data for daily and hourly streamflow simulations utilizing meteorological and runoff data with resolutions ranging from 1 hour to 24 hours from 7 small-to-medium-scale catchments in northern China. Additionally, the transferability of model parameters across varying data resolutions and computation time steps were investigated. Key findings are summarized as follows:

1) For both daily and hourly streamflow simulations, utilizing sub-daily resolution rainfall and streamflow data leads to substantial improvements in model performance compared with the using of the daily data. However, further enhancements in data resolution yield diminishing returns. Specifically, for daily streamflow simulations, improvements in model performance become negligible when the resolution exceeds 12 hours. As for hourly streamflow simulations, improvements in overall flood process accuracy become negligible when the resolution exceeds 6 hours, while higher resolutions further enhance the precision of peak flow predictions.

2) When the model's computation time step is fixed at 1h, most parameters, are generally independent of the input data resolution. Even when using model parameters obtained from daily data, utilizing sub-

daily resolution data helps improve the accuracy of hourly streamflow simulations. Conversely, when the computation time step varies, the model parameters are not applicable for direct transfer to other time steps. In particular, the performance of the model deteriorates more when the computation time step is shifted from large to small.

3) It is recommended to utilized smaller computational time step when constructing hydrological models even in the absence of high-resolution input data. This strategy ensures that the same prediction accuracy is achieved while preserving the transferability of model parameters, thus enhancing the robustness of the model.

**5.2 Limitations and further research needs**

While this study has provided valuable insights into the impacts of data temporal resolution and computational time step on hydrological models, several limitations should be acknowledged. First, this study focuses on a specific geographical area in Northern China and covers a limited temporal range. The findings, therefore, may not be fully generalizable to other regions with different climatic, hydrological, or geological conditions. Further studies across various regions and under different hydrological conditions are necessary to validate and extend the applicability of these results. Second, the study's conclusions are drawn based on a particular hydrological model and specific parameter settings. Other models or configurations might exhibit different sensitivities to data resolution and computational time step. Therefore, the generalization of these findings to other hydrological models should be approached with caution. Next, results showed that the benefit of high-resolution rainfall/streamflow data to daily and hourly streamflow simulation was negligible when the temporal resolution was higher than a threshold, and the possible mechanism of such phenomenon was primarily discussed according to the variation of runoff process and some parameters under different conditions and other existing literatures. However, a deeper analysis and validation on such threshold effect are still lacking, which needs further investigation. Last, the number of iterations for the optimization algorithm during the model calibration process was limited. Although our previous modeling and calibration practices (e.g., Nan and Tian, 2024a, 2024b) demonstrated that the current number of iterations is sufficient to produce a good simulation, it does not guarantee the discovery of a globally optimal result. Consequently, it is challenging to determine whether the slight decline in model performance in certain catchments is due to the high-resolution data or the influence of local optima.

*Competing interests*. At least one of the (co)-authors is a member of the editorial board of Hydrology and Earth System Sciences.

*Acknowledgements.* This study was made possible through the generous financial support from the National Natural Science Foundation of China (grant no. 52309024 and U2442201) and the Fund Program of State Key Laboratory of Hydroscience and Engineering (grant no. sklhseTD-2024-C01).

*Code and data availability*. The data and the code of the model used in this study are available by contacting the authors.

*Author contributions*. YN and FT conceived the idea; MT and YN conducted analysis; FT provided comments on the analysis; and all the authors contributed to writing and revisions.

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
