# Peer review of "Assesing the Value of High-Resolution Data and Parameters Transferability"

_EGUsphere, 2024_

## Referee Comment (RC1)

The paper introduced a novel four-source hydrological model and discussed the value of high-resolution data in hydrological simulations as well as the transferability of model parameters under different conditions. The content is comprehensive, addressing key issues that are critical in hydrological modeling. The writing is clear, well-organized, and the methods are appropriate. Thereby, I recommend the manuscript for acceptance after addressing the following issues.

Below, I outline the issues that I believe need to be resolved or further clarified, which could help improve the quality of the paper.

1. The authors have also investigated the value of high-resolution data for hydrological simulations in another manuscript submitted to this journal. I believe it is necessary to further elaborate on the background, objectives, and the similarities and differences in content and results between this study and the previous one. This will provide readers with a deeper understanding of the topics addressed in this paper.

2. According to Figure 1, it seems that there are some nested relations among the seven catchments, which are not clearly showed by the figure. Besides, it seems that BHP and ZJF belong to the same catchment, and the BHP station is not mentioned in the main text or Table 1. The methodology used to address such nested catchments should be clarified more clearly.

3. In lines 305–306, the authors mention " In all catchments except for XTZ, when parameters calibrated with a specific data resolution were transferred to other resolutions, simulation accuracy improved as the resolution of the data used increased." However, the reasons behind the observed anomalies in this catchment have not been explained. Although the manuscript includes a discussion of another watershed anomaly, I suggest that the authors also provide some explanation or discussion of the observed anomalous behavior in the XTZ catchment to improve the comprehensiveness of the analysis.

---

## Author Comment (AC1)

Thank you for your thorough review and professional feedback on our manuscript. We greatly appreciate your praise for our work, as well as your constructive comments highlighting areas for improvement. We have carefully considered each of your suggestions, and in response, we will revise the manuscript point by point, as per your recommendations, to enhance the overall quality of the paper.

Below is our point-by-point response to your comments and our detailed plans for revision.

**Comment 1:**

The authors have also investigated the value of high-resolution data for hydrological simulations in another manuscript submitted to this journal. I believe it is necessary to further elaborate on the background, objectives, and the similarities and differences in content and results between this study and the previous one. This will provide readers with a deeper understanding of the topics addressed in this paper.

Response:

Thank you for your suggestion. We recognize that our current description of the objectives and novelty of the study is insufficient. This research extends our previous work, with two main objectives: first, to explore the impact of data resolution on hydrological simulations across different climate regions and models, aiming to derive generalizable patterns; and second, to further investigate the transferability of model parameters across different time scales, based on our understanding of the effects of high-resolution data on model performance.

The impact of high-resolution data on hydrological model performance remains unclear, with studies yielding inconsistent results. While some research suggests improvements with higher temporal resolution (Kobold & Brilly, 2006; Jeong et al., 2011), others, like Huang et al. (2019), found only marginal effects from increased spatial resolution, and some studies (Kannan et al., 2006; Ficchì et al., 2016) even reported performance degradation at finer time scales. Our previous research in southern China showed that high-resolution data does not always improve model performance. Nevertheless, we and other related studies acknowledged that further studies across different climate regions and models are necessary to validate and extend the generality of these findings. This research builds upon this background, aiming to provide new perspectives and data for this field.

We believe that the novelty of this study lies in the following aspects:

1. Evaluating the value of high-resolution data in a new climatic region. We specifically chose northern China as the study area because its climate and runoff generation characteristics differ significantly from those of southern China (Fan et al., 2019; Domrös et al., 2012), which we had previously focused on. Furthermore, other studies exploring the impact of data resolution have not yet considered this kind of climatic region.

2. New findings on the impact of data resolution on hydrological modeling in Northern China. Compared to prior studies, our findings reveal that in northern Chinese catchments, increasing data resolution has a more pronounced effect on reducing peak flow errors. Specifically, for hourly flow simulations, NSE (Nash-Sutcliffe Efficiency) showed no significant improvement when the resolution exceeded 6 hours, whereas REP (Relative Error of Peak Flow) only ceased to improve

significantly when the resolution exceeded 3 hours. In southern Chinese catchments, however, the threshold for both NSE and REP was 6 hours. This difference likely arises from variations in runoff-generation characteristics between catchments in different climatic zones.

3. Investigation on transferability of model parameters across various data resolutions and computational time steps. As you pointed out, discovering that model parameters are not transferable when computational time steps change is one of the key novel aspects of this study. Here, we have firstly investigated the value of high-resolution data using a novel model in a new climatic region. The results confirm that our findings align broadly with previous studies and those of other researchers. This supports the generality of the conclusion that "improvements in simulation accuracy become negligible once data resolution surpasses a certain threshold." Building on this understanding, we further broadened the scope of the study to explore the transferability of model parameters, aiming to provide guidance on selecting appropriate computational time steps in environments with lower data resolution. Specifically, we examined the transferability of model parameters under varying data resolutions and computational time steps. The results indicate that parameters remain transferable with changes in data resolution but lose this property when computational time steps change. Based on this finding, we recommend that even in the absence of high-resolution data, hydrological models should be constructed and calibrated using smaller computational time steps whenever possible.

In the revised manuscript, we will update the wording in the introduction and discussion sections to provide readers with a deeper understanding of the topics addressed in this paper.

**Comment 2:**

Figure 1 shows that among the seven catchments studied in this paper, two are not independent. The THK station is located upstream of the ZJF station, and there is also an upstream BHP station, which is not mentioned in the main text. The methodology used to address such nested catchments should be clarified in the manuscript.

Response:

Thank you for pointing out the issue. We neglected to introduce the approach for handling this nested watershed. The Baihe watershed contains four hydrological stations: Xitaizi (XTZ), Tanghekou (THK), Zhangjiafen (ZJF), and Baihepu (BHP). ZJF is the outlet of the Baihe River, while XTZ is the outlet of the small experimental catchment of Xitaizi, and its influence on the flow at ZJF is negligible. THK is located at the outlet of the Tanghe River, a tributary of the Baihe River. Since the flow data for THK is relatively complete, the upstream basin of THK is treated as an independent study catchment. There is a reservoir within the Baihe watershed, called the Baihepu Reservoir. To eliminate human interference and the accumulation of simulation errors, this study treats the catchment area between the Baihepu Reservoir, THK, and ZJF as an independent catchment. The measured outflow from the Baihepu Reservoir and the measured flow at THK are used as known boundary conditions in the hydrological model to simulate the flow at ZJF.

In the revised manuscript, we will include the following description:

Considering the presence of the Baihepu (BHP) Reservoir in the Baihe watershed, and to exclude

human interference and the accumulation of simulation errors, this study treats the catchment area between the Baihepu Reservoir, THK, and ZJF as an independent catchment. The measured outflow from the Baihepu Reservoir and the measured flow at THK are used as known boundary conditions in the hydrological model to simulate the flow at ZJF.

**Comment 3:**

In lines 305–306, the authors mention " In all catchments except for XTZ, when parameters calibrated with a specific data resolution were transferred to other resolutions, simulation accuracy improved as the resolution of the data used increased." However, the reasons behind the observed anomalies in this catchment have not been explained. Although the manuscript includes a discussion of another watershed anomaly, I suggest that the authors also provide some explanation or discussion of the observed anomalous behavior in the XTZ catchment to improve the comprehensiveness of the analysis.

Response:

Thank you for your suggestion. In the original manuscript, we only explained the observed anomalies using one catchment as an example. Similar anomalies were also observed at other sites (such as THK and ZJF), but these anomalies appeared when the data used for model calibration had a resolution of 24h or 12h. In the case of the XTZ catchment, however, these anomalies were evident across most time resolutions.

We analyzed the variation in model parameters with respect to the resolution of the data used for each catchment and found that the majority of parameters did not show significant changes with varying data resolutions, which supports the transferability of the parameters. However, some parameters showed notable changes, which can explain the differences in model performance after parameter transfer. Our analysis revealed that parameters related to the convergence and routing process, such as Lag1 (the lag time for surface runoff) and Muskingum parameters, exhibited considerable variability with changes in time resolution. Among these, the variation in Lag1 was most pronounced at sites like XTZ, THK, and ZJF, with values changing within the 5h-15h range. In contrast, for other catchments, this parameter was generally below 5 hours. This suggests that, despite the relatively small size of the XTZ catchment, its dense vegetation and high infiltration capacity result in a slow rainfall-runoff response. In such catchments, even when using coarser time-resolution data, the model can yield good NSE values (though REP may be poor). This anomaly is consistent with the explanation we provided for the THK site in the original manuscript. We will emphasize the explanation for the observed anomaly in the XTZ catchment in the revised version of the manuscript.

**Reference:**

Domrös M, Peng G. The climate of China[M]. Springer Science & Business Media, 2012.

Fan J, Wu L, Zhang F, et al. Evaluation and development of empirical models for estimating daily and monthly mean daily diffuse horizontal solar radiation for different climatic regions of China[J]. Renewable and Sustainable Energy Reviews, 2019, 105: 168-186.

Huang Y, András Bárdossy, Zhang K .Sensitivity of hydrological models to temporal and spatial resolutions of rainfall data[J].Hydrology and Earth System Sciences, 2019, 23(6):2647-2663.DOI:10.5194/hess-23-2647-2019.

Jeong J, Kannan N, Arnold J G, et al. Development of sub-daily erosion and sediment transport algorithms for SWAT[J]. Transactions of the ASABE, 2011, 54(5): 1685-1691.

Kobold, M. and Brilly, M.: The use of HBV model for flash flood forecasting, Nat. Hazards Earth Syst. Sci., 6, 407–417, https://doi.org/10.5194/nhess-6-407-2006, 2006.

Ficchì A, Perrin C, Andréassian V. Impact of temporal resolution of inputs on hydrological model performance: An analysis based on 2400 flood events[J]. Journal of hydrology, 2016, 538: 454-470.

Kannan N, White S M, Fred W, et al. Sensitivity analysis and identification of the best evapotranspiration and runoff options for hydrological modelling in SWAT-2000 [J]. Journal of Hydrology, 2006,332(3/4):456-466.

---

## Author Comment (AC2)

We greatly appreciate your thorough review and professional comments on our paper. Thank you for recognizing the intent and significance of our work, while also pointing out the current shortcomings of our manuscript, and providing targeted constructive suggestions. We have carefully considered each of your comments, and we will adopt all of your recommendations. The paper will be revised accordingly, point by point, to further enhance its quality.

Below is our point-by-point response to your comments and our detailed plans for revision.

**Comment 1:**

This paper aims to study the transferability of calibrated parameters across temporal scales in hydrological modeling using a single model applied to a specific region in China. These limitations, the narrow focus on a specific region and model, restrict the study's scope. Nevertheless, the vision and idea behind the work are valid and highly relevant.

Response:

Thank you for your comments. We fully understand your concern on the narrow scope of the study. Indeed, this research was conducted on a specific model and region. However, this may not limit the generality of our findings; on the contrary, it is exactly one of the main objectives of our study. We aim to validate and expand the generality of existing conclusions through new research under new specific conditions.

As mentioned in the introduction section, the quantitative benefits of high-resolution data in enhancing hydrological model performance remain unclear. Studies on the impact of data resolution on hydrological models have produced inconsistent results. Kobold and Brilly (2006) found that calibrating hydrological models with sub-daily data and time steps can significantly improve the accuracy of flood forecasting. Similarly, Jeong et al. (2011) observed similar improvements. Huang et al. (2019) found that increasing spatial resolution has only a marginal or minimal effect on model performance, while high temporal resolution data leads to a significant improvement in model performance. However, other studies (Kannan et al., 2006; Ficchì et al., 2016) have found that higher data resolution does not always lead to better model performance. Ficchì et al. (2016) reported that as the time scale is reduced, the improvement in model performance becomes limited, and performance may even degrade. Our previous research (Tudaji et al., 2024) in southern China showed that high-resolution data does not always have positive impact on model performance. Nevertheless, we and other related studies acknowledged that further studies across different climate regions and models are necessary to validate and extend the generality of these findings. In fact, the marginal effect of data resolution on model performance is expected. The focus should be on exploring the specific values of the threshold resolution, the underlying causes of performance degradation, and their generalizability across different climate regions and models. This research builds upon this background, aiming to provide new perspectives and data for this field.

We specifically chose northern China as the study area because its climate and runoff generation characteristics differ significantly from those of southern China, which we had previously focused on (Fan et al., 2019; Domrös et al., 2012). Furthermore, other studies exploring the impact of data resolution have not yet considered northern China. Southern China features subtropical and tropical

monsoon climates, with warm, humid conditions and abundant, evenly distributed rainfall. Annual precipitation typically exceeds 800 mm (averaging 1500 mm in our study area), classifying it as a humid region. Flood generation is predominantly governed by saturation excess. In contrast, northern China experiences a temperate monsoon climate with lower and more concentrated rainfall. Annual precipitation is generally below 800 mm (averaging 600 mm in our study area), making it a semi-humid to semi-arid region where flood generation is primarily driven by infiltration excess and subsurface preferential flow. By applying a new model to this distinct region, we aim to further validate the previous findings and investigate the role of high-resolution data under different climatic conditions.

In addition, we have broadened the scope of our study by incorporating an investigation into the transferability of model parameters. This aims to provide guidance on using existing parameters across different time scales or selecting appropriate computational time steps in conditions with lack of high-resolution data. Although the model we employed differs in certain aspects, its general structure and computational approaches are similar to those of most hydrological models. For example, it uses linear methods to calculate subsurface runoff and the Muskingum method for routing. Therefore, our findings are applicable to this category of hydrological models and offer valuable insights for practitioners utilizing such models.

We acknowledge that the original manuscript did not adequately explain the background and objectives of this study, nor did it effectively integrate our findings with existing literature. This may have made the study's focus appear overly narrow. In the revised manuscript, we will elaborate on the background and objectives, and discuss the general patterns of how high-resolution data influence hydrological simulations, drawing on findings from studies conducted in different climate zones and with different models.

**Comment 2:**

The authors should better highlight the novelties of this work compared to the recent study, https://doi.org/10.5194/egusphere-2024-1438. As a consequence, all parts related to model performance across temporal scales should be downplayed, as they currently lack sufficient depth and originality.
The most significant novelty lies in the finding that model parameters are not transferable when the computational time step varies. However, the observation that higher-resolution data produces better results has already been extensively discussed, with numerous examples provided in the ongoing work https://doi.org/10.5194/egusphere-2024-1438, which is openly accessible in the HESS discussion section.

Response:

Thank you for your insightful suggestions. We recognize the inadequacy of not highlighting the novelty of our study. As mentioned earlier, we did not sufficiently elaborate on the progress of existing research and the background of this study, which resulted in the novelty of our work being underrepresented. We believe that the novelty of this study lies in the following aspects:

1. Evaluating the value of high-resolution data in a new climatic region. This study assesses the

value of high-resolution data for hydrological simulations in a new climatic zone, aiming to validate the generality of conclusions drawn from previous studies. The climatic characteristics of the study area have been introduced above. We will add this introduction in the revised manuscript.

2. New findings on the impact of data resolution on hydrological modeling in Northern China. Compared to prior studies, our findings reveal that in northern Chinese catchments, increasing data resolution has a more pronounced effect on reducing peak flow errors. Specifically, for hourly flow simulations, NSE (Nash-Sutcliffe Efficiency) showed no significant improvement when the resolution exceeded 6 hours, whereas REP (Relative Error of Peak Flow) only ceased to improve significantly when the resolution exceeded 3 hours. In southern Chinese catchments, however, the threshold for both NSE and REP was 6 hours. This difference likely arises from variations in runoff-generation characteristics between catchments in different climatic zones.

3. Investigation on transferability of model parameters across various data resolutions and computational time steps. As you pointed out, discovering that model parameters are not transferable when computational time steps change is one of the key novel aspects of this study. Here, we have firstly investigated the value of high-resolution data using a novel model in a new climatic region. The results confirm that our findings align broadly with previous studies and those of other researchers. This supports the generality of the conclusion that "improvements in simulation accuracy become negligible once data resolution surpasses a certain threshold." Building on this understanding, we further broadened the scope of the study to explore the transferability of model parameters, aiming to provide guidance on selecting appropriate computational time steps in environments with lower data resolution. Specifically, we examined the transferability of model parameters under varying data resolutions and computational time steps. The results indicate that parameters remain transferable with changes in data resolution but lose this property when computational time steps change. Based on this finding, we recommend that even in the absence of high-resolution data, hydrological models should be constructed and calibrated using smaller computational time steps whenever possible.

In the revised manuscript, we will update the wording in the introduction and discussion sections to emphasize these points of novelty.

**Comment 3:**
HESS primarily focuses on generalizable findings rather than region-specific studies. Despite differences in the study areas, the aims of both works, if understood correctly, appear to focus on deriving general conclusions. However, the current discussion does not sufficiently support this objective, making the paper feel more suited to journals focused on regional case studies.

Response:

One of the objectives of this study is precisely to identify generalizable patterns. To this end, we intentionally selected a new climatic zone for our research, complementing previous studies. Although this research was conducted in a specific region using a specific model, our study area represents a typical climatic zone. While the specific equations used in our model differ, its structure and certain assumptions are similar to those of most mainstream hydrological models. Therefore, the methods and conclusions of this study are supposed to be generalizable. Certainly, we need to

thoroughly integrate the conclusions of our study with existing research findings.

The title, "A case study," may lead to a misunderstanding that this research is merely a case study of a specific region. If the editor allows, we would prefer to revise the title in the revised manuscript to avoid this confusion.

**Comment 4:**

Furthermore, the model is not adequately described. The lack of detailed model descriptions significantly limits the reader's ability to understand the factors that might influence the transferability of parameters across temporal scales. For instance, mechanisms like snow dynamics or evapotranspiration processes could provide valuable insights into why certain parameters can or cannot be transferred. Similarly, a discussion of potential model limitations, perhaps in the modeling of snow dynamics or evapotranspiration, could clarify the conditions under which the model performs better or worse. Unfortunately, these aspects are overlooked, leaving the paper as a lengthy description with insufficient critical evaluation or insight.

Response:

Thank you for pointing out this shortcoming. We now recognize the drawback of not providing a detailed description of the model used in the study. In the methodology section, we focused on describing the experimental design, while the specific equations used in the model were placed in the supplementary section, which led to insufficient explanation of the model itself. Just as you illustrated with examples, our model includes linear assumptions in some modules, such as using a linear reservoir to calculate groundwater runoff, and incorporating lag time in surface runoff calculations. The values of these parameters are related to the time step used, which results in their non-transferability. Although we discussed some variations of these parameters in Section 4.2, we now realize that without a proper explanation of the model, this discussion may not be clearly conveyed. In the revised manuscript, we will provide a detailed description of the model and further specifically elaborate on how the parameter changes with time scales and its (non-)transferability in Sections 4.2 and 4.3, in conjunction with the formula and significance where the parameter is located.

Here is the description of the model that we intend to include in Section 3.1 of the revised manuscript.

[revised manuscript text omitted]

---

## Author Response (AR1)

Dear editor and reviewers,

Thank you for your thorough review of our manuscript. We sincerely appreciate your comments and constructive suggestions, which helped us improve the quality of our study. We have carefully considered all of your comments and revised the manuscript. The following are point-to-point response to each comment including all relevant changes made in the manuscript.

**Editor**

(Both reviewers acknowledged some strengths of this manuscript, but the major concerns from both reviewers and myself are)

**Comment 1:** Better highlight the novelties of this work compared to their recent study, https://doi.org/10.5194/egusphere-2024-1438.

**Response:** Thank you for your suggestion. We recognize that our current description of the objectives and novelty of the study is insufficient. The first comment from the first reviewer and the second comment from the second reviewer also emphasized the necessity of making this revision. We have responded carefully to each in turn (please see our responses to them below for details).

To better highlight the novelties of this work, we updated the wording in the introduction and discussion sections (Lines 42-51, 57-64, 81-88, 420-439), and added a new subsection (Lines 566-608) in the Discussion section on the transferability of parameters across different calculation time steps.

**Comment 2:** The most significant novelty lies in the finding that model parameters are not transferable when the computational time step varies. The authors should provide better explanations for the observed anomalous behavior.

**Response:** Thanks for your suggestion. Following your and the reviewers' suggestions, we have first added a detailed description of the model's structure and its computational methods in Section 2.1, so that readers can better understand the role of model parameters in the simulation. We then identified the sensitive parameters and confirmed that the routing process is the dominant factor determining the transferability of parameters. We quantified the impact of the transfer of the Muskingum parameters on model performance and provided the trends of these parameters across different time scales. The above has been added as a separate section (Section 4.3) to better explain the non- transferability of parameters across computational time steps.

**Comment 3:** The generalization of the findings from this region-specific studies.
**Response:** The second reviewer has also expressed concerns regarding this issue in his first and third comment. We have addressed these concerns with careful consideration (please see our detailed responses in the corresponding sections below).
Though this study focuses on a specific region and model, the climate type of the study area in this study is not uncommon, and the structure and algorithms of the hydrological model used are similar

to those of mainstream models. This ensures the generalizability and applicability of the conclusions within a certain range.

We acknowledge that the original manuscript lacked sufficient description of the background, objectives, study area, and the model used, which may have led to difficulties for readers in understanding the conditions under which the study's conclusions were drawn, thereby raising concerns on their generalizability. In response, we have revised the manuscript by updating the wording in the introduction and discussion sections, and have added more detailed descriptions of the study area and model structure (Lines 85-88, 119, 154-208, 427-439). These revisions aim to clearly outline the conditions under which our conclusions were derived, as well as the climate regions and model types to which these conclusions are applicable.

**Comment 4:** The model is not adequately described. The lack of detailed model descriptions significantly limits the reader's ability to understand the factors that might influence the transferability of parameters across temporal scales. In addition, this paper has also insufficient critical evaluation or insight.

**Response:** Thank you for pointing out this shortcoming. We recognize the drawback of not providing a detailed description of the model used in the study. In the methodology section, we focused on describing the experimental design, while the specific equations used in the model were placed in the supplementary section, which led to insufficient explanation of the model itself. we have added a detailed description of the model to Section 3.1 of the revised manuscript.

This study leads to the following main conclusions: 1) Increasing the time resolution of data does not always result in significant improvements in model performance when using a semi-distributed hydrological model; 2) While model parameters are transferable across different data resolutions, they are not transferable when using different computational time steps. These conclusions are further supported by analyses of the distribution of the model's optimal parameters. Based on these findings, we offer insightful recommendations for hydrological model development in the absence of high-resolution data. Contrary to traditional belief, which suggests that accurate flow simulation at a fine time scale must require high-resolution data, our findings indicate that the required data resolution for modeling does not necessarily need to exceed the resolution of the target data. Even in the absence of high-resolution input data, we recommend using smaller computational time steps when constructing hydrological models. This approach ensures the transferability of model parameters across different resolutions, and when higher-resolution data becomes available in the future, these parameters remain applicable. As a result, this strategy enhances the robustness of the model while maintaining its predictive accuracy.

**Reviewer #1**

**Comment 1:**

The authors have also investigated the value of high-resolution data for hydrological simulations in another manuscript submitted to this journal. I believe it is necessary to further elaborate on the background, objectives, and the similarities and differences in content and results between this study and the previous one. This will provide readers with a deeper understanding of the topics addressed in this paper.

Response:

Thank you for your suggestion. We recognize that our current description of the objectives and novelty of the study is insufficient. This research extends our previous work, with two main objectives: first, to further investigate the value of high-resolution data in hydrological simulations using a new model in a different climate region, aiming to derive generalizable patterns; and second, to further investigate the transferability of model parameters across different time scales, based on our understanding of the effects of high-resolution data on model performance.

The impact of high-resolution data on hydrological model performance remains unclear, with studies yielding inconsistent results. While some research suggests improvements with higher temporal resolution (Kobold & Brilly, 2006; Jeong et al., 2011), others, like Huang et al. (2019), found only marginal effects from increased spatial resolution, and some studies (Kannan et al., 2006; Ficchì et al., 2016) even reported performance degradation at finer time scales. Our previous research in southern China showed that high-resolution data does not always improve model performance. Nevertheless, we and other related studies acknowledged that further studies across different climate regions and models are necessary to validate and extend the generality of these findings. This research builds upon this background, aiming to provide new perspectives and data for this field.

The novelty of this study lies in the following aspects:

1. Evaluating the value of high-resolution data in a new climatic region. We specifically chose northern China as the study area because its climate and runoff generation characteristics differ significantly from those of southern China (Fan et al., 2019; Domrös et al., 2012), which we had previously focused on. Additionally, to our knowledge, other studies investigating the impact of data resolution have not yet considered this climatic region.

2. New findings on the impact of data resolution on hydrological modeling in Northern China. Compared to prior studies, our findings reveal that in northern Chinese catchments, increasing data resolution has a more pronounced effect on reducing peak flow errors. Specifically, for hourly flow simulations, NSE (Nash-Sutcliffe Efficiency) showed no significant improvement when the resolution exceeded 6 hours, whereas REP (Relative Error of Peak Flow) only ceased to improve significantly when the resolution reached and exceeded 2 hours. In southern Chinese catchments, however, the threshold for both NSE and REP was 6 hours. This difference arises from variations in runoff-generation characteristics between catchments in different climatic zones. In our revised manuscript, we have added this discussion.

3. Investigation on transferability of model parameters across various data resolutions and

computational time steps. Discovering that model parameters are not transferable when computational time steps change is one of the key novel aspects of this study. Here, we have firstly investigated the value of high-resolution data using a novel model in a new climatic region. The results confirm that our findings align broadly with previous study and those of other researchers. This supports the generality of the conclusion that "improvements in simulation accuracy become negligible once data resolution surpasses a certain threshold." Building on this understanding, we further broadened the scope of the study to explore the transferability of model parameters, aiming to provide guidance on selecting appropriate computational time steps in conditions with lower data resolution. Specifically, we examined the transferability of model parameters under varying data resolutions and computational time steps. In this section of the investigation, we provide a detailed description of the trends in sensitive parameters with respect to time scale, aiming to explain the impact of data resolution on hydrological simulations from the perspective of the model simulation process, and to explore the main causes of model parameter transferability or non-transferability. The results indicate that parameters remain transferable with changes in data resolution but lose this property when computational time steps change. Based on this finding, we recommend that even in the absence of high-resolution data, hydrological models should be constructed and calibrated using smaller computational time steps whenever possible.

In the revised manuscript, we updated the wording in the introduction and discussion sections (Lines 42-51, 57-64, 81-88, 420-439), and added a new subsection (Lines 566-608) in the Discussion section on the transferability of parameters across different calculation time steps to provide readers with a deeper understanding of the topics addressed in this paper.

**Comment 2:**

Figure 1 shows that among the seven catchments studied in this paper, two are not independent. The THK station is located upstream of the ZJF station, and there is also an upstream BHP station, which is not mentioned in the main text. The methodology used to address such nested catchments should be clarified in the manuscript.

Response:

Thank you for pointing out the issue. We neglected to introduce the approach for handling this nested watershed. The Baihe watershed contains four hydrological stations: Xitaizi (XTZ), Tanghekou (THK), Zhangjiafen (ZJF), and Baihepu (BHP). ZJF is the outlet of the Baihe River, while XTZ is the outlet of the small experimental catchment of Xitaizi, and its influence on the flow at ZJF is negligible. THK is located at the outlet of the Tanghe River, a tributary of the Baihe River. Since the flow data for THK is relatively complete, the upstream basin of THK is treated as an independent study catchment. There is a reservoir within the Baihe watershed, called the Baihepu Reservoir. To eliminate human interference and the accumulation of simulation errors, this study treats the catchment area between the Baihepu Reservoir, THK, and ZJF as an independent catchment. The measured outflow from the Baihepu Reservoir and the measured flow at THK are used as known boundary conditions in the hydrological model to simulate the flow at ZJF.

Lines 114-119 of the revised manuscript contain the additions we made in response to the above-mentioned issue.

**Comment 3:**

In lines 305–306, the authors mention " In all catchments except for XTZ, when parameters calibrated with a specific data resolution were transferred to other resolutions, simulation accuracy improved as the resolution of the data used increased." However, the reasons behind the observed anomalies in this catchment have not been explained. Although the manuscript includes a discussion of another watershed anomaly, I suggest that the authors also provide some explanation or discussion of the observed anomalous behavior in the XTZ catchment to improve the comprehensiveness of the analysis.

Response:

Thank you for your suggestion. In the original manuscript, we only explained the observed anomalies using one catchment as an example. Similar anomalies were also observed at other sites (such as THK and ZJF), but these anomalies appeared when the data used for model calibration had a resolution of 24h or 12h. In the case of the XTZ catchment, however, these anomalies were evident across most time resolutions.

We analyzed the variation in model parameters with respect to the resolution of the data used for each catchment and found that the majority of parameters did not show significant changes with varying data resolutions, which supports the transferability of the parameters. However, some parameters showed notable changes, which can explain the differences in model performance after parameter transfer. Our analysis revealed that parameters related to the convergence and routing process, such as Lag1 (the lag time for surface runoff) and Muskingum parameters, exhibited considerable variability with changes in time resolution. Among these, the variation in Lag1 was most pronounced at sites like XTZ, THK, and ZJF, with values changing within the 5h-15h range. In contrast, for other catchments, this parameter was generally below 5 hours. The substantial divergence in the optimal parameters for models employing low-resolution versus high-resolution data results in a decline in model efficacy. This decline occurs even when high-resolution data is employed, provided that the parameters are those optimized for low-resolution data scenarios. This provides an explanation for why the NSE at XTZ, THK, and ZJF stations exhibits a decline when using high-resolution data, whereas other stations do not experience such a decrease.

We have added the above explanation in the revised version of the manuscript (Lines 531-538).

**Reference:**

Domrös M, Peng G. The climate of China[M]. Springer Science & Business Media, 2012.

Fan J, Wu L, Zhang F, et al. Evaluation and development of empirical models for estimating daily and monthly mean daily diffuse horizontal solar radiation for different climatic regions of China[J]. Renewable and Sustainable Energy Reviews, 2019, 105: 168-186.

Huang Y, András Bárdossy, Zhang K .Sensitivity of hydrological models to temporal and spatial resolutions of rainfall data[J].Hydrology and Earth System Sciences, 2019, 23(6):2647-2663.DOI:10.5194/hess-23-2647-2019.

Jeong J, Kannan N, Arnold J G, et al. Development of sub-daily erosion and sediment transport

algorithms for SWAT[J]. Transactions of the ASABE, 2011, 54(5): 1685-1691.

Kobold, M. and Brilly, M.: The use of HBV model for flash flood forecasting, Nat. Hazards Earth Syst. Sci., 6, 407–417, https://doi.org/10.5194/nhess-6-407-2006, 2006.

Ficchì A, Perrin C, Andréassian V. Impact of temporal resolution of inputs on hydrological model performance: An analysis based on 2400 flood events[J]. Journal of hydrology, 2016, 538: 454-470.

Kannan N, White S M, Fred W, et al. Sensitivity analysis and identification of the best evapotranspiration and runoff options for hydrological modelling in SWAT-2000 [J]. Journal of Hydrology, 2006,332(3/4):456-466.

**Reviewer #2**

**Comment 1:**

This paper aims to study the transferability of calibrated parameters across temporal scales in hydrological modeling using a single model applied to a specific region in China. These limitations, the narrow focus on a specific region and model, restrict the study's scope. Nevertheless, the vision and idea behind the work are valid and highly relevant.

Response:
Thank you for your constructive comments. We appreciate your recognition of the validity and relevance of the vision and idea behind this study. We fully understand your concern regarding the narrow scope of the study. Indeed, this research focuses on a specific model and region, which limits the generalizability of the findings. However, this focused approach is, in fact, a key part of our research objectives. By studying new specific conditions in a different climate region, we aim to test and refine existing conclusions, expanding their applicability. We acknowledge that studies focusing on specific regions may not lead to universally applicable conclusions. Therefore, we intend to address this gap by conducting research in different regions and in collaboration with other researchers, ultimately contributing to the development of more generalizable conclusions. Besides, the climate type of the study area in this study is not uncommon, and the structure and algorithms of the hydrological model used are similar to those of mainstream models. This ensures the generalizability and applicability of the conclusions within a certain range.

As mentioned in the introduction section, the quantitative benefits of high-resolution data in enhancing hydrological model performance remain unclear. Studies on the impact of data resolution on hydrological models have produced inconsistent results. Kobold and Brilly (2006) found that calibrating hydrological models with sub-daily data and time steps can significantly improve the accuracy of flood forecasting. Similarly, Jeong et al. (2011) observed similar improvements. Huang et al. (2019) found that increasing spatial resolution has only a marginal or minimal effect on model performance, while high temporal resolution data leads to a significant improvement in model performance. However, other studies (Kannan et al., 2006; Ficchì et al., 2016) have found that higher data resolution does not always lead to better model performance. Ficchì et al. (2016) reported that as the time scale is reduced, the improvement in model performance becomes limited, and performance may even degrade. Our previous research (Tudaji et al., 2024) in southern China showed that high-resolution data does not always have positive impact on model performance. Nevertheless, we and other related studies acknowledged that further studies across different climate regions and models are necessary to validate and extend the generality of these findings. In fact, the marginal effect of data resolution on model performance is expected. The focus should be on exploring the specific values of the threshold resolution, the underlying causes of performance degradation, and their generalizability across different climate regions and models. This research builds upon this background, aiming to provide new perspectives and data for this field.

We specifically chose northern China as the study area because its climate and runoff generation characteristics differ significantly from those of southern China (Fan et al., 2019; Domrös et al., 2012), which we had previously focused on. Furthermore, other studies exploring the impact of data resolution have not yet considered northern China. Southern China features subtropical and tropical monsoon climates, with warm, humid conditions and abundant, evenly distributed rainfall. Annual

precipitation typically exceeds 800 mm (averaging 1500 mm in our previous study area), classifying it as a humid region. Flood generation is predominantly governed by saturation excess. In contrast, northern China experiences a temperate monsoon climate with lower and more concentrated rainfall. Annual precipitation is generally below 800 mm (averaging 600 mm in our study area), making it a semi-humid to semi-arid region where flood generation is primarily driven by infiltration excess and subsurface preferential flow. By applying a new model to this distinct region, we aim to further validate the previous findings and investigate the role of high-resolution data under different climatic conditions. This was also confirmed by the results of the study, which showed that impacts of high-resolution data on the simulation of peak flow in the catchments in the two climate regions above are different.

In addition, we have broadened the scope of our study by incorporating an investigation into the transferability of model parameters. This aims to provide guidance on using existing parameters across different time scales or selecting appropriate computational time steps in conditions with lack of high-resolution data. Although the model we employed differs in certain aspects, its general structure and computational approaches are similar to those of most hydrological models. For example, it uses a lag algorithm for surface runoff and the Muskingum method for channel routing, both of which are widely used in many mainstream hydrological models. This study specifically focuses on the transferability of parameters associated with these methods. Therefore, our findings are applicable to this category of hydrological models and offer valuable insights for practitioners utilizing such models.

We acknowledge that the original manuscript lacked sufficient description of the background, objectives, study area, and the model used, which may have led to difficulties for readers in understanding the conditions under which the study's conclusions were drawn, thereby raising concerns on their generalizability. In response, we have revised the manuscript by updating the wording in the introduction and discussion sections, and have added more detailed descriptions of the study area and model structure (Lines 85-88, 119, 154-208, 427-439). These revisions aim to clearly outline the conditions under which our conclusions were derived, as well as the climate regions and model types to which these conclusions are applicable.

**Comment 2:**

The authors should better highlight the novelties of this work compared to the recent study, https://doi.org/10.5194/egusphere-2024-1438. As a consequence, all parts related to model performance across temporal scales should be downplayed, as they currently lack sufficient depth and originality.

The most significant novelty lies in the finding that model parameters are not transferable when the computational time step varies. However, the observation that higher-resolution data produces better results has already been extensively discussed, with numerous examples provided in the ongoing work https://doi.org/10.5194/egusphere-2024-1438, which is openly accessible in the HESS discussion section.

Response:

Thank you for your insightful suggestions. We recognize the inadequacy of not highlighting the

novelty of our study. As mentioned earlier, we did not sufficiently elaborate on the progress of existing research and the background of this study, which resulted in the novelty of our work being underrepresented. The novelty of this study lies in the following aspects:

1. Evaluating the value of high-resolution data in a new climatic region. We specifically chose northern China as the study area because its climate and runoff generation characteristics differ significantly from those of southern China (Fan et al., 2019; Domrös et al., 2012), which we had previously focused on. Additionally, to our knowledge, other studies investigating the impact of data resolution have not yet considered this climatic region.

2. New findings on the impact of data resolution on hydrological modeling in Northern China. Compared to prior studies, our findings reveal that in northern Chinese catchments, increasing data resolution has a more pronounced effect on reducing peak flow errors. Specifically, for hourly flow simulations, NSE (Nash-Sutcliffe Efficiency) showed no significant improvement when the resolution exceeded 6 hours, whereas REP (Relative Error of Peak Flow) only ceased to improve significantly when the resolution reached and exceeded 2 hours. In southern Chinese catchments, however, the threshold for both NSE and REP was 6 hours. This difference arises from variations in runoff-generation characteristics between catchments in different climatic zones. In our revised manuscript, we have added this discussion.

3. Investigation on transferability of model parameters across various data resolutions and computational time steps. Discovering that model parameters are not transferable when computational time steps change is one of the key novel aspects of this study. Here, we have firstly investigated the value of high-resolution data using a novel model in a new climatic region. The results confirm that our findings align broadly with previous study and those of other researchers. This supports the generality of the conclusion that "improvements in simulation accuracy become negligible once data resolution surpasses a certain threshold." Building on this understanding, we further broadened the scope of the study to explore the transferability of model parameters, aiming to provide guidance on selecting appropriate computational time steps in conditions with lower data resolution. Specifically, we examined the transferability of model parameters under varying data resolutions and computational time steps. In this section of the investigation, we provide a detailed description of the trends in sensitive parameters with respect to time scale, aiming to explain the impact of data resolution on hydrological simulations from the perspective of the model simulation process, and to explore the main causes of model parameter transferability or non-transferability. The results indicate that parameters remain transferable with changes in data resolution but lose this property when computational time steps change. Based on this finding, we recommend that even in the absence of high-resolution data, hydrological models should be constructed and calibrated using smaller computational time steps whenever possible.

In the revised manuscript, we have updated the wording in the introduction and discussion sections. Additionally, we introduced a new subsection (lines 566-608) in the Discussion section addressing the transferability of parameters across different computational time steps. In this subsection, we identified the sensitive parameters and quantified their impact on model performance, providing the trends of these parameters across different time scales. These update and addition aim to better explain why model parameters may not be transferable when the computational time step changes, highlighting the novelty of this study relative to other studies.

**Comment 3:**

HESS primarily focuses on generalizable findings rather than region-specific studies. Despite differences in the study areas, the aims of both works, if understood correctly, appear to focus on deriving general conclusions. However, the current discussion does not sufficiently support this objective, making the paper feel more suited to journals focused on regional case studies.

Response:

We fully understand your concern regarding the generality of the study. By studying new specific conditions in a different climate region, we aim to test and refine existing conclusions, expanding their applicability. We acknowledge that studies focusing on specific regions may not lead to universally applicable conclusions. Therefore, we intend to address this gap by conducting research in different regions, ultimately contributing to the development of more generalizable conclusions.

Besides, the climate type of the selected study area in this study is not uncommon, and the structure and algorithms of the hydrological model used are similar to those of mainstream models. This ensures the generalizability and applicability of the conclusions within a certain range.

In the revised manuscript, we have updated the wording in the introduction and added a more detailed description of the study area and the model used, to help readers better understand the objectives and purpose of this study, as well as the conditions under which the conclusions were drawn and their applicability. Through this, we aim to ensure that this study, like other case studies published in HESS, provides valuable references for similar research, thereby contributing to the development of more broadly generalizable conclusions in the field.

**Comment 4:**

Furthermore, the model is not adequately described. The lack of detailed model descriptions significantly limits the reader's ability to understand the factors that might influence the transferability of parameters across temporal scales. For instance, mechanisms like snow dynamics or evapotranspiration processes could provide valuable insights into why certain parameters can or cannot be transferred. Similarly, a discussion of potential model limitations, perhaps in the modeling of snow dynamics or evapotranspiration, could clarify the conditions under which the model performs better or worse. Unfortunately, these aspects are overlooked, leaving the paper as a lengthy description with insufficient critical evaluation or insight.

Response:

Thank you for pointing out this shortcoming. We recognize the drawback of not providing a detailed description of the model used in the study. In the methodology section, we focused on describing the experimental design, while the specific equations used in the model were placed in the supplementary section, which led to insufficient explanation of the model itself. we have added a detailed description of the model to Section 3.1 of the revised manuscript.

As you pointed out, a specific hydrological process module of the model may be the dominant factor determining whether parameters can be transferred. Following your suggestion, we identified sensitive parameters and confirmed that the routing process is the dominant factor. In the revised

manuscript, we have quantified the impact of the Muskingum parameters' transfer on model performance and provided the trends of these parameters across different time scales. A detailed discussion on this has been added to the revised manuscript in lines 566-608.

**Reference:**

Cunge, J., 1969. On the subject of a flood propagation computation method (Muskingum method). Journal of Hydraulic Research, 7, 205–230.

McCarthy, G.T., 1938. The unit hydrograph and flood routing. In Proceedings of the Conference of North Atlantic Division, US Engineer Department, New London, CN, 608–609.

Domrös M, Peng G. The climate of China[M]. Springer Science & Business Media, 2012.

Fan J, Wu L, Zhang F, et al. Evaluation and development of empirical models for estimating daily and monthly mean daily diffuse horizontal solar radiation for different climatic regions of China[J]. Renewable and Sustainable Energy Reviews, 2019, 105: 168-186.

Huang Y, András Bárdossy, Zhang K .Sensitivity of hydrological models to temporal and spatial resolutions of rainfall data[J].Hydrology and Earth System Sciences, 2019, 23(6):2647-2663.DOI:10.5194/hess-23-2647-2019.

Jeong J, Kannan N, Arnold J G, et al. Development of sub-daily erosion and sediment transport algorithms for SWAT[J]. Transactions of the ASABE, 2011, 54(5): 1685-1691.

Kobold, M. and Brilly, M.: The use of HBV model for flash flood forecasting, Nat. Hazards Earth Syst. Sci., 6, 407–417, https://doi.org/10.5194/nhess-6-407-2006, 2006.

Ficchì A, Perrin C, Andréassian V. Impact of temporal resolution of inputs on hydrological model performance: An analysis based on 2400 flood events[J]. Journal of hydrology, 2016, 538: 454-470.

Kannan N, White S M, Fred W, et al. Sensitivity analysis and identification of the best evapotranspiration and runoff options for hydrological modelling in SWAT-2000 [J]. Journal of Hydrology, 2006,332(3/4):456-466.

Zhao, R. J. (1992). The Xinanjiang model applied in China. Journal ofHydrology,135(1-4),371-381.

Yu Y, Song X, Zhang Y, et al. Impact of reclaimed water in the watercourse of Huai River on groundwater from Chaobai River basin, Northern China[J]. Frontiers of earth science, 2017, 11: 643-659.

Cunge, J., 1969. On the subject of a flood propagation computation method (Muskingum method). Journal of Hydraulic Research, 7, 205–230.

McCarthy, G.T., 1938. The unit hydrograph and flood routing. In Proceedings of the Conference of North Atlantic Division, US Engineer Department, New London, CN, 608–609.